# Gradient-Free Minimax Optimization: Variance Reduction and Faster Convergence

## Abstract

Many important machine learning applications amount to solving minimax optimization problems, and in many cases there is no access to the gradient information, but only the function values. In this paper, we focus on such a gradient-free setting, and consider the nonconvex-strongly-concave minimax stochastic optimization problem. In the literature, various zeroth-order (i.e., gradient-free) minimax methods have been proposed, but none of them achieve the potentially feasible computational complexity of $\mathcal{O}(\epsilon^{-3})$ suggested by the stochastic nonconvex minimization theorem. In this paper, we adopt the variance reduction technique to design a novel zeroth-order variance reduced gradient descent ascent (ZO-VRGDA) algorithm. We show that the ZO-VRGDA algorithm achieves the best known query complexity of $\mathcal{O}(\kappa(d_1 + d_2)\epsilon^{-3})$, which outperforms all previous complexity bounds by orders of magnitude, where $d_1$ and $d_2$ denote the dimensions of the optimization variables and $\kappa$ denotes the condition number. In particular, with a new analysis technique that we develop, our result does not rely on a diminishing or accuracy-dependent stepsize usually required in the existing methods. To our best knowledge, this is the first study of zeroth-order minimax optimization with variance reduction. Experimental results on the black-box distributional robust optimization problem demonstrates the advantageous performance of our new algorithm.[1]

## 1 Introduction

Minimax optimization has attracted significant growth of attention in machine learning as it captures several important machine learning models and problems including generative adversarial networks (GANs) Goodfellow et al. (2014), robust adversarial machine learning Madry et al. (2018), imitation learning Ho & Ermon (2016), etc. Minimax optimization typically takes the following form:

$$\min_{x \in \mathbb{R}^{d_1}} \max_{y \in \mathbb{R}^{d_2}} f(x, y), \text{where } f(x, y) \triangleq \begin{cases} \mathbb{E}[F(x, y; \xi)] & \text{(online case)} \\ \frac{1}{n} \sum_{i=1}^{n} F(x, y; \xi_i) & \text{(finite-sum case)} \end{cases} \tag{1}$$

where $f(x, y)$ takes the expectation form if data samples $\xi$ are taken in an online fashion, and $f(x, y)$ takes the finite-sum form if a dataset of training samples $\xi_i$ for $i = 1, \ldots, n$ are given in advance.

This paper focuses on the nonconvex-strongly-concave minimax problem, in which $f(x, y)$ is nonconvex with respect to $x$ for all $y \in \mathbb{R}^{d_2}$, and $f(x, y)$ is $\mu$-strongly concave with respect to $y$ for all $x \in \mathbb{R}^{d_1}$. The problem then takes the following equivalent form:

$$\min_{x \in \mathbb{R}^{d_1}} \left\{ \Phi(x) \triangleq \max_{y \in \mathbb{R}^{d_2}} f(x, y) \right\}, \tag{2}$$

where the objective function $\Phi(\cdot)$ in eq. (2) is nonconvex in general.

In many machine learning scenarios, minimax optimization problems need to be solved without access to the gradient information, but only to the function values, e.g., in multi-agent reinforcement learning with bandit

---

[1]This paper was initially posted on arXiv in June 2020.

Table 1: Comparison of gradient-free algorithms for nonconvex-strongly-concave minimax problems

| Algorithm | Estimator | Stepsize | Overall Complexity |
|---|---|---|---|
| ZO-min-max Liu et al. (2019) | UniGE | $\mathcal{O}(\kappa^{-1}\ell^{-1})$ | $\mathcal{O}((d\epsilon^{-6})$ |
| ZO-SGDA Wang et al. (2020) | GauGE | $\mathcal{O}(\kappa^{-4}\ell^{-1})$ | $\mathcal{O}(d\kappa^5\epsilon^{-4})$ |
| ZO-SGDMSA Wang et al. (2020) | GauGE | $\mathcal{O}(\kappa^{-1}\ell^{-1})$ | $\mathcal{O}(d\kappa^2\epsilon^{-4}\log(\frac{1}{\epsilon}))$ |
| ZO-VRGDA (this work) | GauGE | $\mathcal{O}(\kappa^{-1}\ell^{-1})$ | $\mathcal{O}(d\kappa^3\epsilon^{-3})$ |

[1] "UniGE" and "GauGE" stand for "Uniform smoothing Gradient Estimator" and "Gaussian smoothing Gradient Estimator", respectively.

[2] The complexity refers to the total number of queries of the function value.

[3] We include only the complexity in the online case in the table, because many previous studies did not consider the finite-sum case. We comment on the finite-sum case in Section 4.

[4] We define $d = d_1 + d_2$.

feedback Wei et al. (2017); Zhang et al. (2019) and robotics Wang & Jegelka (2017); Bogunovic et al. (2018). Such scenarios have motivated the design of **gradient-free (i.e., zeroth-order)** algorithms, which solve the problem by querying the function values. For nonconvex-strongly-concave minimax optimization, stochastic gradient descent (SGD) type algorithms have been proposed, which use function values to form gradient estimators in order to iteratively find the solution. In particular, Liu et al. (2019) studied a constrained problem and proposed a ZO-min-max algorithm that achieves an $\epsilon$-accurate solution with the function query complexity of $\mathcal{O}((d_1 + d_2)\epsilon^{-6})$. Wang et al. (2020) designed ZO-SGDA and ZO-SGDMSA, and between the two algorithms ZO-SGDMA achieves the better function query complexity of $\mathcal{O}((d_1 + d_2)\kappa^2\epsilon^{-4}\log(1/\epsilon))$.

Despite the previous progress, if we view the minimax problem as the nonconvex problem in eq. (2), the lower bound on the computational complexity suggests that zeroth-order algorithms may potentially achieve the query complexity of $\mathcal{O}((d_1 + d_2)\epsilon^{-3})$. But none of the previous algorithms in the literature achieves such a desirable rate. Thus, a fundamental question to ask here is as follows.

- *Can we design a better gradient-free algorithm that outperforms all existing stochastic algorithms by orders of magnitude, and can achieve the desired query complexity of $\mathcal{O}((d_1 + d_2)\epsilon^{-3})$ suggested by the lower bound of gradient-based algorithms?*

This paper provides an affirmative answer to the above question together with the development of novel analysis tools.

## 1.1 Main Contributions

We propose the first zeroth-order variance reduced gradient descent ascent (ZO-VRGDA) algorithm for minimax optimization. ZO-VRGDA features gradient-free designs and adopts a nested-loop structure with the recursive variance reduction method incorporated for both the inner- and outer-loop updates. In particular, the outer loop adopts zeroth-order coordinate-wise estimators for accurate gradient estimation, and the inner loop adopts zeroth-order Gaussian smooth estimators for efficient gradient estimation. This is the first gradient-free variance reduced algorithm designed for minimax optimization.

We establish the convergence rate and the function query complexity for ZO-VRGDA for nonconvex-strongly-concave achieves the best known query complexity of $\mathcal{O}((d_1 + d_2)\kappa^3\epsilon^{-3})$, which outperforms the existing state-of-the-art (achieved by ZO-SGDMSA Wang et al. (2020)) in the case with $\epsilon \leq \kappa^{-1}$. For the finite-sum case, we show that ZO-VRGDA achieves an overall query complexity of $\mathcal{O}((d_1 + d_2)(\kappa^2\sqrt{n}\epsilon^{-2} + n) + d_2(\kappa^2 + \kappa n)\log(\kappa))$ when $n \geq \kappa^2$, and $\mathcal{O}((d_1 + d_2)(\kappa^2 + \kappa n)\kappa\epsilon^{-2})$ when $n \leq \kappa^2$. Our work provides the first convergence analysis for gradient-free variance reduced algorithms for minimax optimization.

It is also instructive to compare our result with a concurrent work Huang et al. (2020), which proposed an accelerated zeroth-order momentum descent ascent (Acc-ZOMDA) method for minimax optimization.

The performance difference between our ZO-VRGDA and their Acc-ZOMDA is two folds. (a) The query complexity of our ZO-VRGDA outperforms that of Acc-ZOMDA by a factor of $(d_1 + d_2)^{1/2}$, which can be significant in large dimensional problems such as the neural network training. (b) Rigorously speaking, our result characterizes the exact convergence to an $\epsilon$-accurate stationary point, whereas the convergence metric in Huang et al. (2020) does not necessarily imply convergence to a stationary point.

From the technical standpoint, differently from the previous approach (e.g., Luo et al. (2020)), we develop a new analysis framework for analyzing recursive variance reduced algorithms for minimax problems. Specifically, the main challenge for our analysis lies in bounding two inter-connected stochastic error processes: tracking error and gradient estimation error. The previous analysis forces those two error terms to be kept at $\epsilon$-level at the cost of inefficient initialization and $\epsilon$-level small stepsize. In contrast, we develop new tools to capture the coupling of the accumulative estimation error and tracking error over the entire algorithm execution, and then establish their relationships with the accumulative gradient estimators to derive an overall convergence bound. As a result, our ZO-VRGDA can adopt a more relaxed initialization and a large constant stepsize for fast running speed, and still enjoy the theoretical convergence guarantee.

## 1.2   Related Work

Due to the vast number of studies on minimax optimization and on variance reduced algorithms, we include below only the studies that are most relevant to this work.

Variance reduction methods for minimax optimization are inspired by those for conventional minimization problems, including SAGA Defazio et al. (2014); Reddi et al. (2016), SVRG Johnson & Zhang (2013); Allen-Zhu & Hazan (2016); Allen-Zhu (2017), SARAH Nguyen et al. (2017a;b; 2018), SPIDER Fang et al. (2018), SpiderBoost Wang et al. (2019), etc. But the convergence analysis for minimax optimization is much more challenging, and is typically quite different from its counterparts in minimization problems.

For *strongly-convex-strongly-concave minimax optimization*, Palaniappan & Bach (2016) applied SVRG and SAGA to the finite-sum case and established a linear convergence rate, and Chavdarova et al. (2019) proposed SVRE later to obtain a better bound. When the condition number of the problem is very large, Luo et al. (2019) proposed a proximal point iteration algorithm to improve the performance of SAGA. For some special cases, Du et al. (2017); Du & Hu (2019) showed that the linear convergence rate of SVRG can be maintained without the strongly-convex or strongly concave assumption. Yang et al. (2020) applied SVRG to study the minimax optimization under the two-sided Polyak-Lojasiewicz condition.

*Nonconvex-strongly-concave minimax optimization* is the focus of this paper. As we discuss at the beginning of the introduction, SGD-type algorithms have been developed and studied, including SGDmax Jin et al. (2019), PGSMD Rafique et al. (2018), and SGDA Lin et al. (2019). Xu et al. (2021) proposed alternative zeroth-order GDA algorithms for solving both general smooth and block-wise nonsmooth nonconvex-concave minimax problems. Chen et al. (2021b); Yang et al. (2020; 2022) established a global optimality guarantee for GDA algorithms in nonconvex minimax optimization under special landscape assumptions. Chen et al. (2021a) developed a cubic-regularized GDA algorithm for nonconvex minimax optimization, which is guaranteed to escape the sub-optimal points. Several variance reduction methods have also been proposed to further improve the performance, including PGSVRG Rafique et al. (2018), the SAGA-type algorithm for minimax optimization Wai et al. (2019), and SREDA Luo et al. (2020). Particularly, SREDA has been shown in Luo et al. (2020) to achieve the optimal complexity dependence on $\epsilon$.

While SGD-type zeroth-order algorithms have been studied for minimax optimization, such as Menickelly & Wild (2020); Roy et al. (2019) for convex-concave minimax problems and Liu et al. (2019); Wang et al. (2020) for nonconvex-strongly-concave minimax problems, variance reduced algorithms have not been developed for *zeroth-order minimax optimization* so far. This paper proposes the first such algorithm named ZO-VRGDA for nonconvex-strongly-concave minimax optimization, and established its complexity performance which outperforms the existing comparable algorithms (see Table 1).

### 1.3 Notation

In this paper, we use $\left\|\cdot\right\|_2$ to denote the Euclidean norm of vectors. For a finite set $\mathcal{S}$, we denote its cardinality as $\left|\mathcal{S}\right|$. For a positive integer $n$, we denote $[n] = \{1, \cdots, n\}$.

## 2 Preliminaries

We first introduce the gradient estimator that we use to design our gradient-free algorithm, and then describe the technical assumptions that we take in our analysis.

### 2.1 Zeroth-order Gradient Estimator

We consider the Gaussian smoothed function Nesterov & Spokoiny (2017); Ghadimi & Lan (2013) defined as

$$f_{\mu_1}(x, y) := \mathbb{E}_{\nu, \xi} F(x + \mu_1 \nu, y, \xi),$$
$$f_{\mu_2}(x, y) := \mathbb{E}_{\omega, \xi} F(x, y + \mu_2 \omega, \xi),$$

where $\nu_i \sim N(0, \mathbf{1}_{d_1})$, $\omega_i \sim N(0, \mathbf{1}_{d_2})$ with $\mathbf{1}_d$ denoting the identity matrices with sizes $d \times d$. Then, in order to approximate the gradient of $f_{\mu_1}(x, y)$ and $f_{\mu_2}(x, y)$ with respect to $x$ and $y$ based on the function values, the zeroth-order stochastic gradient estimators can be constructed as

$$G_{\mu_1}(x, y, \nu_{\mathcal{M}_1}, \xi_{\mathcal{M}}) = \frac{1}{|\mathcal{M}|} \sum_{i \in [|\mathcal{M}|]} \frac{F(x + \mu_1 \nu_i, y, \xi_i) - F(x, y, \xi_i)}{\mu_1} \nu_i, \tag{3}$$

$$H_{\mu_2}(x, y, \omega_{\mathcal{M}_2}, \xi_{\mathcal{M}}) = \frac{1}{|\mathcal{M}|} \sum_{i \in [|\mathcal{M}|]} \frac{F(x, y + \mu_2 \omega_i, \xi_i) - F(x, y, \xi_i)}{\mu_2} \omega_i, \tag{4}$$

where $|\mathcal{M}| = |\mathcal{M}_1| = |\mathcal{M}_2|$ denote the batchsize of samples. It can be shown that $G_{\mu_1}(x, y, \nu_{\mathcal{M}_1}, \xi_{\mathcal{M}})$ and $H_{\mu_2}(x, y, \omega_{\mathcal{M}_2}, \xi_{\mathcal{M}})$ are unbiased estimators of the true gradient of $f_{\mu_1}(x, y)$ and $f_{\mu_2}(x, y)$ with respect to $x$ and $y$ Ghadimi & Lan (2013), respectively, i.e.,

$$\mathbb{E}_{\nu_{\mathcal{M}_1}, \xi_{\mathcal{M}}} G_{\mu_1}(x, y, \nu_{\mathcal{M}_1}, \xi_{\mathcal{M}}) = \nabla_x f_{\mu_1}(x, y),$$
$$\mathbb{E}_{\omega_{\mathcal{M}_2}, \xi_{\mathcal{M}}} H_{\mu_2}(x, y, \omega_{\mathcal{M}_2}, \xi_{\mathcal{M}}) = \nabla_y f_{\mu_2}(x, y).$$

These zeroth-order gradient estimators are useful for us to design a gradient-free algorithm for minimax optimization.

### 2.2 Technical Assumptions

We take the following standard assumptions for the minimax problem in eq. (1) or eq. (2), which have also been adopted in Liu et al. (2019); Wang et al. (2020); Huang et al. (2020); Luo et al. (2020); Lin et al. (2019). We slightly abuse the notation $\xi$ below to represent the random index in both the online and finite-sum cases, where in the finite-sum case, $\mathbb{E}_\xi[\cdot]$ is with respect to the uniform distribution over $\{\xi_1, \cdots, \xi_n\}$.

**Assumption 1.** *The function $\Phi(\cdot)$ is lower bounded, i.e., we have $\Phi^* = \inf_{x \in \mathbb{R}^{d_1}} \Phi(x) > -\infty$.*

**Assumption 2.** *The component function $F$ has an averaged $\ell$-Lipschitz gradient, i.e., for all $(x, y), (x', y') \in \mathbb{R}^{d_1} \times \mathbb{R}^{d_2}$, we have $\mathbb{E}_\xi \left[ \left\|\nabla F(x, y; \xi) - \nabla F(x', y'; \xi)\right\|_2^2 \right] \le \ell^2 (\left\|x - x'\right\|_2^2 + \left\|y - y'\right\|_2^2)$.*

**Assumption 3.** *The function $f$ is $\mu$-strongly-concave in $y$ for any $x \in \mathbb{R}^{d_1}$, and the component function $F$ is concave in $y$, i.e., for any $x \in \mathbb{R}^{d_1}$, $y, y' \in \mathbb{R}^{d_2}$ and $\xi$, we have*

$$f(x, y) \le f(x, y') + \langle \nabla_y f(x, y'), y - y' \rangle - \frac{\mu}{2} \left\|y - y'\right\|_2,$$

*and*

$$F(x, y; \xi) \le F(x, y'; \xi) + \langle \nabla_y F(x, y'; \xi), y - y' \rangle.$$

---

**Algorithm 1** ZO-VRGDA

---

1: **Input:** $x_0$, initial accuracy $\zeta$, learning rate $\alpha = \Theta(\frac{1}{\kappa \ell})$, $\beta = \Theta(\frac{1}{\ell})$, batch size $\mathcal{S}_1$, $\mathcal{S}_2$ and periods $q, m$.
2: **Initialization:** $y_0 = $ ZO-iSARAH$(-f(x_0, \cdot), \zeta)$ (detailed in Algorithm 2)
3: **for** $t = 0, 1, ..., T - 1$ **do**
4:     **if** $\mathrm{mod}(k, q) = 0$ **then**
5:         draw $S_1$ samples $\{\xi_1, \cdots, \xi_{S_1}\}$
6:         $v_t = \frac{1}{S_1} \sum_{i=1}^{S_1} \sum_{j=1}^{d_1} \frac{F(x_t + \delta e_j, y_t, \xi_i) - F(x_t - \delta e_j, y_t, \xi_i)}{2\delta} e_j$
7:         $u_t = \frac{1}{S_1} \sum_{i=1}^{S_1} \sum_{j=1}^{d_2} \frac{F(x_t, y_t + \delta e_j, \xi_i) - F(x_t, y_t - \delta e_j, \xi_i)}{2\delta} e_j$
8:         where $e_j$ denotes the vector with $j$-th natural unit basis vector.
9:     **else**
10:         $v_t = \tilde{v}_{t-1, \bar{m}_{t-1}}$, $u_t = \tilde{u}_{t-1, \bar{m}_{t-1}}$
11:     **end if**
12:     $x_{t+1} = x_t - \alpha v_t$
13:     $y_{t+1} = $ ZO-ConcaveMaximizer$(t, m, S_{2,x}, S_{2,y})$ (detailed in Algorithm 3)
14: **end for**
15: **Output:** $\hat{x}$ chosen uniformly at random from $\{x_t\}_{t=0}^{T-1}$

---

**Assumption 4.** *The gradient of each component function $F(x, y; \xi)$ has a bounded variance, i.e., there exists a constant $\sigma > 0$ such that for any $(x, y) \in \mathbb{R}^{d_1 \times d_2}$, we have*

$$\mathbb{E}_\xi \big[ \|\nabla F(x, y; \xi) - \nabla f(x, y)\|_2^2 \big] \leq \sigma^2.$$

Note that the above variance assumption is weaker than that of Acc-ZOMDA in Huang et al. (2020), because Huang et al. (2020) directly requires the variance of the zeroth-order estimator to be bounded, which is not easy to verify. In contrast, we require such a condition to hold only for the original stochastic gradient estimator, which is standard in the optimization literature and can be satisfied easily in practice.

We define $\kappa \triangleq \ell / \mu$ as the condition number of the problem throughout the paper. The following structural lemma developed in Lin et al. (2019) provides further information about $\Phi$ for nonconvex-strongly-concave minimax optimization.

**Lemma 1** (Lemma 3.3 of Lin et al. (2019))**.** *Under Assumption 2 and 3, the function $\Phi(\cdot) = \max_{y \in \mathbb{R}^{d_2}} f(\cdot, y)$ is $(\kappa + 1)\ell$-gradient Lipschitz and $\nabla \Phi(x) = \nabla_x f(x, y^*(x))$ is $\kappa$-Lipschitz, where $y^*(\cdot) = \mathrm{argmin}_{y \in \mathbb{R}^{d_2}} f(\cdot, y)$.*

We let $L \triangleq (1 + \kappa)\ell$ denote the Lipschitz constant of $\nabla \Phi(x)$. Since $\Phi$ is nonconvex in general, it is NP-hard to find its global minimum. Our goal here is to develop a gradient-free zeroth-order stochastic gradient algorithms that output an $\epsilon$-stationary point as defined below.

**Definition 1.** *The point $\bar{x}$ is called an $\epsilon$-stationary point of the differentiable function $\Phi$ if $\|\nabla \Phi(\bar{x})\|_2 \leq \epsilon$, where $\epsilon$ is a positive constant.*

## 3 ZO-VRGDA: Zeroth-Order Variance Reduction Algorithm

In this section, we propose a new zeroth-order variance reduced gradient descent ascent (ZO-VRGDA) algorithm to solve the minimax problem in eq. (1) or eq. (2). ZO-VRGDA (see Algorithm 1) adopts a nested-loop structure, in which the parameters $x_t$ and $y_t$ are updated in a nested loop fashion: each update of $x_t$ in the outer-loop is followed by $(m + 1)$ updates of $y_t$ over one entire inner loop. ZO-VRGDA incorporates the variance reduction method for both the inner-loop and outer-loop updates, and features gradient-free designs. We next describe the ZO-VRGDA algorithm in more detail as follows.

(a) The initialization of ZO-VRGDA (line 2 of Algorithm 1) utilizes a zeroth-order algorithm ZO-iSARAH (see Algorithm 2), which adopts a first-order algorithm iSARAH and incorporates the zeroth-order gradient estimators, to search an initialization $y_0$ with predefined accuracy $\mathbb{E}[\|\nabla_y f(x_0, y_0)\|_2^2] \leq \zeta$. In particular, ZO-iSARAH uses a small batch of sampled function values to construct Gaussian estimators for approximating gradients (line 10 of Algorithm 2), which is defined as

---

**Algorithm 2** ZO-iSARAH

---

1: **Input:** $\tilde{w}_0$, learning rate $\gamma > 0$, inner loop size $I$, batch size $B_1$ and $B_2$
2: **for** $t = 1, 2, ..., T$ **do**
3:     $w_0 = \tilde{w}_{t-1}$
4:     draw $B_1$ samples $\{\xi_1, \cdots, \xi_{B_1}\}$
5:     $v_0 = \frac{1}{B_1} \sum_{i=1}^{B_1} \sum_{j=1}^{d} \frac{P(w_0 + \delta e_j, \xi_i) - P(w_0 - \delta e_j, \xi_i)}{2\delta} e_j$
6:     where $e_j$ denotes the vector with $j$-th natural unit basis vector.
7:     $w_1 = w_0 + \gamma v_0$
8:     **for** $k = 1, 2, ..., I - 1$ **do**
9:         Draw minibatch sample $\mathcal{M} = \{\xi_1, \cdots, \xi_{B_2}\}$ and $\mathcal{M}_1 = \{\psi_1, \cdots, \psi_{B_2}\}$
10:        $v_k = v_{k-1} + \Psi_\tau(w_k, \psi_{\mathcal{M}_1}, \xi_{\mathcal{M}}) - \Psi_\tau(w_{k-1}, \psi_{\mathcal{M}_1}, \xi_{\mathcal{M}})$
11:        $w_{k+1} = w_k - \gamma v_k$
12:     **end for**
13:     $\tilde{w}_t$ chosen uniformly at random from $\{w_k\}_{k=0}^{I}$
14: **end for**

---

**Algorithm 3** ZO-ConcaveMaximizer$(t, m, S_{2,x}, S_{2,y})$

---

1: **Initialization:** $\tilde{x}_{t,-1} = x_t$, $\tilde{y}_{t,-1} = y_t$, $\tilde{x}_{t,0} = x_{t+1}$, $\tilde{y}_{t,0} = y_t$, $\tilde{v}_{t,-1} = v_t$, $\tilde{u}_{t,-1} = u_t$
2: Draw minibatch sample $\mathcal{M}_x = \{\xi_1, \cdots, \xi_{S_{2,x}}\}$, $\mathcal{M}_{1,x} = \{\nu_1, \cdots, \nu_{S_{2,x}}\}$ and $\mathcal{M}_{2,x} = \{\omega_1, \cdots, \omega_{S_{2,x}}\}$, and $\mathcal{M}_y = \{\xi_1, \cdots, \xi_{S_{2,y}}\}$, $\mathcal{M}_{1,x} = \{\nu_1, \cdots, \nu_{S_{2,y}}\}$ and $\mathcal{M}_{2,y} = \{\omega_1, \cdots, \omega_{S_{2,y}}\}$
3: $\tilde{v}_{t,0} = \tilde{v}_{t,-1} + G(\tilde{x}_{t,0}, \tilde{y}_{t,0}, \nu_{\mathcal{M}_{1,x}}, \xi_{\mathcal{M}_x}) - G(\tilde{x}_{t,-1}, \tilde{y}_{t,-1}, \nu_{\mathcal{M}_{1,x}}, \xi_{\mathcal{M}_x})$
4: $\tilde{u}_{t,0} = \tilde{u}_{t,-1} + H(\tilde{x}_{t,0}, \tilde{y}_{t,0}, \omega_{\mathcal{M}_{2,y}}, \xi_{\mathcal{M}_y}) - H(\tilde{x}_{t,-1}, \tilde{y}_{t,-1}, \omega_{\mathcal{M}_{2,y}}, \xi_{\mathcal{M}_y})$
5: $\tilde{x}_{t,1} = \tilde{x}_{t,0}$
6: $\tilde{y}_{t,1} = \tilde{y}_{t,0} + \beta \tilde{u}_{t,0}$
7: **for** $k = 1, 2, ..., m + 1$ **do**
8:     Draw minibatch sample $\mathcal{M}_x = \{\xi_1, \cdots, \xi_{S_{2,x}}\}$, $\mathcal{M}_{1,x} = \{\nu_1, \cdots, \nu_{S_{2,x}}\}$ and $\mathcal{M}_{2,x} = \{\omega_1, \cdots, \omega_{S_{2,x}}\}$, and $\mathcal{M}_y = \{\xi_1, \cdots, \xi_{S_{2,y}}\}$, $\mathcal{M}_{1,y} = \{\nu_1, \cdots, \nu_{S_{2,y}}\}$ and $\mathcal{M}_{2,y} = \{\omega_1, \cdots, \omega_{S_{2,y}}\}$
9:     $\tilde{v}_{t,k} = \tilde{v}_{t,k-1} + G_{\mu_1}(\tilde{x}_{t,k}, \tilde{y}_{t,k}, \nu_{\mathcal{M}_{1,x}}, \xi_{\mathcal{M}_x}) - G_{\mu_1}(\tilde{x}_{t,k-1}, \tilde{y}_{t,k-1}, \nu_{\mathcal{M}_{1,x}}, \xi_{\mathcal{M}_x})$
10:    $\tilde{u}_{t,k} = \tilde{u}_{t,k-1} + H_{\mu_2}(\tilde{x}_{t,k}, \tilde{y}_{t,k}, \omega_{\mathcal{M}_{2,y}}, \xi_{\mathcal{M}_y}) - H_{\mu_2}(\tilde{x}_{t,k-1}, \tilde{y}_{t,k-1}, \omega_{\mathcal{M}_{2,y}}, \xi_{\mathcal{M}_y})$
11:    $\tilde{x}_{t,k+1} = \tilde{x}_{t,k}$
12:    $\tilde{y}_{t,k+1} = \tilde{y}_{t,k} + \beta \tilde{u}_{t,k}$
13: **end for**
**output** $y_{t+1} = \tilde{y}_{t,\tilde{m}_t}$ with $\tilde{m}_t$ chosen uniformly at random from $\{0, 1, \cdots, m\}$

---

$$\Psi_\tau(w, \psi_{\mathcal{M}_1}, \xi_{\mathcal{M}}) = \frac{1}{|\mathcal{M}|} \sum_{i \in [|\mathcal{M}|]} \frac{P(w + \tau \psi_i, \xi_i) - P(w, \xi_i)}{\tau} \psi_i, \tag{5}$$

where $\psi_i \sim N(0, \mathbf{1}_d)$.

(b) The outer-loop updates of $x_t$ is divided into epochs for variance reduction. Consider a certain outer-loop epoch $t = \{(n_t - 1)q, \cdots, n_t q - 1\}$ ($1 \leq n_t < \lceil T/q \rceil$ is a positive integer). At the beginning of such an epoch, ZO-VRGDA utilizes a large batch $S_1$ of the sampled function values to construct gradient-free coordinate-wise estimators for gradient $\nabla_x f(x, y)$ and $\nabla_y f(x, y)$ (see lines 6 and 7 in Algorithm 1). Note that the coordinate-wise gradient estimator is commonly taken in the zeroth-order variance reduced algorithms such as in Ji et al. (2019); Fang et al. (2018) for minimization problems. The batch size $S_1$ is set to be large so that gradient estimators that recursively updated in each epoch can build on an accurate estimators ($v_t$ and $u_t$). In this way, the estimators recursively updated over the entire epoch will not deviate too much from the exact gradients.

(c) For each outer-loop iteration, an inner loop of ZO-ConcaveMaximizer (see Algorithm 3) (line 13 of ZO-VRGDA) uses the small batch $S_{2,x}$ and $S_{2,y}$ of sampled function values to construct a variance reduced estimators for $\nabla_x f_{\mu_1}(x, y)$ and $\nabla_y f_{\mu_2}(x, y)$, respectively, as follows:

$$\tilde{v}_{t,k} = \tilde{v}_{t,k-1} + G_{\mu_1}(\tilde{x}_{t,k}, \tilde{y}_{t,k}, \nu_{\mathcal{M}_{1,x}}, \xi_{\mathcal{M}_x}) - G_{\mu_1}(\tilde{x}_{t,k-1}, \tilde{y}_{t,k-1}, \nu_{\mathcal{M}_{1,x}}, \xi_{\mathcal{M}_x})$$
$$\tilde{u}_{t,k} = \tilde{u}_{t,k-1} + H_{\mu_2}(\tilde{x}_{t,k}, \tilde{y}_{t,k}, \omega_{\mathcal{M}_{2,y}}, \xi_{\mathcal{M}_y}) - H_{\mu_2}(\tilde{x}_{t,k-1}, \tilde{y}_{t,k-1}, \omega_{\mathcal{M}_{2,y}}, \xi_{\mathcal{M}_y}).$$

where the estimators $G_\mu(\cdot)$ and $H_\mu(\cdot)$ are defined in Section 2.1. These zeroth-order gradient estimators are then recursively updated through the inner loop. The batch size $S_2$ is set at the same scale as epoch length $q$, so that the accumulated error of the recursively updated estimators $\tilde{v}_{t,k}$ and $\tilde{u}_{t,k}$ can be kept at a relatively low level.

In addition to the above major gradient-free designs, ZO-VRGDA also features the following enhancements over its first-order counterpart SREDA Luo et al. (2020). (a) ZO-VRGDA relaxes the initialization requirement to be $\mathbb{E}[\|\nabla_y f(x_0, y_0)\|_2^2] \leq \kappa^{-1}$, which requires only $\mathcal{O}(\kappa \log \kappa)$ gradient estimations. This improves the computational cost by a factor of $\tilde{\mathcal{O}}(\kappa\epsilon^{-2})$. (b) ZO-VRGDA adopts a much larger and $\epsilon$-**in**dependent stepsize $\alpha_t = \alpha = \mathcal{O}(1/(\kappa\ell))$ for $x_t$ so that each outer-loop update can make much bigger progress.

# 4 Convergence Analysis of ZO-VRGDA

In this section, we first present our convergence results for ZO-VRGDA and then provide a proof sketch for our analysis.

## 4.1 Main Results

In order to analyze the convergence of ZO-VRGDA, we first provide the complexity analysis for the initialization algorithm ZO-iSARAH. Since the initialization is applied to the variable $y$, with respect to which the objective function is strongly concave. Hence, the initialization is equivalent to the following standard optimization problem:

$$\min_{w \in \mathbb{R}^d} p(w) \triangleq \mathbb{E}[P(w; \xi)], \tag{6}$$

where $P$ is average $\ell$-gradient Lipschitz and convex, $p$ is $\mu$-strongly convex, and $\xi$ is a random vector.

It turns out that the convergence of the zeroth-order recursive variance reduced algorithm ZO-iSARAH has not been studied before for strongly convex optimization. We thus provide the first complexity result for ZO-iSARAH to solve the problem in eq. (6) as follows.

**Theorem 1.** *Apply ZO-iSARAH in Algorithm 2 to solve the strongly convex optimization problem in eq. (6). Set $\gamma = \Theta(1/\ell)$, $B_1 = \Theta(1/\epsilon)$, $B_2 = d$, $I = \Theta(\kappa)$, $T = \Theta(\log(1/\epsilon))$, $\delta = \Theta(\epsilon^{0.5}/\ell d^{0.6})$, and $\tau = \min\{\frac{\epsilon^{0.5}}{3\ell(d+3)^{1.5}}, \sqrt{\frac{2\epsilon}{5\ell\mu d}}\}$. Then, the output of Algorithm 2 satisfies*

$$\mathbb{E}[\|\nabla p_\tau(\tilde{w}_T)\|_2^2] \leq \epsilon,$$

*with the total function query complexity given by*

$$T \cdot (I \cdot B_2 + d \cdot B_1) = \mathcal{O}\left(d\left(\kappa + \frac{1}{\epsilon}\right)\log\left(\frac{1}{\epsilon}\right)\right).$$

Since we require the initialization accuracy in Algorithm 1 to be $\kappa^{-1}$, Theorem 1 indicates that the total function query complexity of performing ZO-iSARAH in Algorithm 1 is $\mathcal{O}(d_2\kappa\log(1/\kappa))$. Ignoring the dependence on the dimension caused by zeroth-order estimator, our initialization complexity improves upon its first-order counterpart SREDA Luo et al. (2020) by a factor of $\tilde{\mathcal{O}}(\kappa\epsilon^{-2})$.

We next provide our main theorem as follows, which characterizes the query complexity of ZO-VRGDA for finding a first-order stationary point of $\Phi(\cdot)$ with $\epsilon$ accuracy.

**Theorem 2.** *Apply ZO-VRGDA in Algorithm 1 to solve the online case of the problem eq. (1). Suppose Assumptions 1-4 hold. Consider the following hyperparamter setting: $\zeta = \kappa^{-1}$, $\alpha = \mathcal{O}(\kappa^{-1}\ell^{-1})$, $\beta = \mathcal{O}(\ell^{-1})$, $q = \mathcal{O}(\epsilon^{-1})$, $m = \mathcal{O}(\kappa)$, $S_1 = \mathcal{O}(\sigma^2\kappa^2\epsilon^{-2})$, $S_{2,x} = \mathcal{O}(d_1\kappa\epsilon^{-1})$, $S_{2,y} = \mathcal{O}(d_2\kappa\epsilon^{-1})$, $\delta = \mathcal{O}((d_1+d_2)^{0.5}\kappa^{-1}\ell^{-1}\epsilon)$, $\mu_1 = \mathcal{O}(d_1^{-1.5}\kappa^{-2.5}\ell^{-1}\epsilon)$ and $\mu_2 = \mathcal{O}(d_2^{-1.5}\kappa^{-2.5}\ell^{-1}\epsilon)$. Then for $T$ to be at least at the order of $\mathcal{O}(\kappa\epsilon^{-2})$, Algorithm 1 outputs $\hat{x}$ such that*

$$\mathbb{E}[\|\nabla\Phi(\hat{x})\|_2] \leq \epsilon,$$

*with the overall function query complexity given by*

$$T \cdot (S_{2,x} + S_{2,y}) \cdot m + \left\lceil \frac{T}{q} \right\rceil \cdot S_1 \cdot (d_1 + d_2) + T_0$$
$$= \mathcal{O}\left( \frac{\kappa}{\epsilon^2} \cdot \frac{(d_1 + d_2)\kappa}{\epsilon} \cdot \kappa \right) + \mathcal{O}\left( \frac{\kappa}{\epsilon} \cdot \frac{\kappa^2}{\epsilon^2} \cdot (d_1 + d_2) \right) + \mathcal{O}\left( d_2 \kappa \log(\kappa) \right)$$
$$= \mathcal{O}\left( (d_1 + d_2)\kappa^3 \epsilon^{-3} \right). \tag{7}$$

Furthermore, ZO-VRGDA can also be applied to the finite-sum case of the problem eq. (1), by replacing the large batch sample $S_1$ used in line 6 of Algorithm 1 with the full set of samples. Then the following result characterizes the query complexity in such a case.

**Theorem 3.** *Apply ZO-VRGDA described above to solve the finite-sum case of the problem eq. (1). Suppose Assumptions 1-4 hold. Under appropriate parameter settings given in Appendix E, the function query complexity to attain an $\epsilon$-stationary point is $\mathcal{O}((d_1 + d_2)(\sqrt{n}\kappa^2 \epsilon^{-2} + n) + d_2(\kappa^2 + \kappa n)\log(\kappa))$ for $n \geq \kappa^2$, and $\mathcal{O}((d_1 + d_2)(\kappa^2 + \kappa n)\epsilon^{-2})$ for $n \leq \kappa^2$.*

Theorem 2 and Theorem 3 indicate that the query complexity of ZO-VRGDA matches the optimal dependence on $\epsilon$ of the first-order algorithm for nonconvex optimization in Fang et al. (2018). The dependence on $d_1$ and $d_2$ typically arises in zeroth-order algorithms due to the estimation of gradients with dimensions $d_1$ and $d_2$. Furthermore, in the online case, ZO-VRGDA outperforms the best known query complexity dependence on $\epsilon$ among the existing zeroth-order algorithms by a factor of $\mathcal{O}(1/\epsilon)$. Including the conditional number $\kappa$ into consideration, ZO-VRGDA outperforms the best known query complexity achieved by ZO-SGDMA in the case with $\epsilon \leq \kappa^{-1}$ (see Table 1).

Theorem 2 and Theorem 3 provide the first convergence analysis and the query complexity for the zeroth-order variance-reduced algorithms for minimax optimization. Furthermore, Theorem 3 provides the first query complexity for the finite-sum zeroth-order minimax problems.

## 4.2 Outline of Technical Proof

Our analysis has the following two major novel developments. (a) We develop new tools to analyze the zeorth-order estimator for variance reduced minimax algorithms. (b) More importantly, differently from the previous approach (e.g., Luo et al. (2020)), we develop a new analysis framework for analyzing the recursive variance reduced algorithms for minimax problems. At a high level, the previous analysis mainly focuses on bounding two inter-related errors: **tracking error** $\delta_t = \mathbb{E}[\|\nabla_y f(x_t, y_t)\|_2^2]$ that captures how well $y_t$ approximates the optimal point $y^*(x_t)$ for a given $x_t$, and **gradient estimation error** $\Delta_t = \mathbb{E}[\|v_t - \nabla_x f(x_t, y_t)\|_2^2 + \|u_t - \nabla_y f(x_t, y_t)\|_2^2]$ that captures how well the stochastic gradient estimators approximate the true gradients. In the previous analysis, those two error terms are forced to be at $\epsilon$-level at the cost of inefficient initialization and $\epsilon$-level stepsize. In contrast, we develop tools to capture the coupling of the accumulative estimation and tracking errors over the entire algorithm execution, and then establish their relationships with the accumulative gradient estimators to derive the overall convergence bound. As a result, our ZO-VRGDA can adopt a more relaxed initialization and a large constant stepsize for fast running speed, and still enjoy a theoretical convergence guarantee.

**Proof Sketch of Theorem 2.** The proof of Theorem 2 consists of the following three steps.

**Step 1:** We start from the estimation error $\Delta_t'$ and tracking error $\delta_t'$ defined with respect to the Gaussian smooth objective functions: $\Delta_t' = \mathbb{E}[\|\nabla_x f_{\mu_1}(x_t, y_t) - v_t\|_2^2] + \mathbb{E}[\|\nabla_y f_{\mu_2}(x_t, y_t) - u_t\|_2^2]$ and $\delta_t' = \mathbb{E}[\|\nabla_y f_{\mu_2}(x_t, y_t)\|_2^2]$. which is connected with $\Delta_t$ and $\delta_t$ via the following inequalities:

$$\Delta_t \leq 2\Delta_t' + \frac{\mu_1^2}{2}\ell^2(d_1 + 3)^3 + \frac{\mu_2^2}{2}\ell^2(d_2 + 3)^3,$$
$$\delta_t \leq 2\delta_t' + \frac{\mu_2^2}{2}\ell^2(d_2 + 3)^3.$$

We establish the relationship between $\Delta'_t$ and $\Delta'_{t-1}$ as well as that between $\delta'_t$ and $\delta'_{t-1}$ as follows:

$$\Delta'_t \leq (1 + \Theta(\epsilon)) \Delta'_{t-1} + \Theta(\epsilon)\delta'_{t-1} + \Theta(\kappa^{-2}\epsilon)\mathbb{E}[\|v_{t-1}\|_2^2] + \Theta(\kappa^{-2}\epsilon^2), \tag{8}$$

$$\delta'_t \leq \frac{1}{2}\delta'_{t-1} + \Theta(1)\Delta'_{t-1} + \Theta(\kappa^{-2})\mathbb{E}[\|v_{t-1}\|_2^2] + \Theta(\kappa^{-5}\epsilon^2). \tag{9}$$

**Step 2: Step 1** indicates that $\Delta'_t$ and $\delta'_t$ are strongly coupled with each other at each iteration. Then, we need to decouple them so that we can characterize the effect of $\Delta'_t$ and $\delta'_t$ on the overall convergence separately.

We first consider the accumulation of $\Delta'_t$ over one epoch. Although the value of $\Delta'_t$ increases within each epoch (indicated by eq. (8)), the accumulation of this error can still be controlled via adjusting the mini-batch sizes $S_1$, $S_2$ and epoch length $q$. Under an appropriate parameter setting, we can obtain the following bound:

$$\Delta'_t \leq 2\Delta'_{\lfloor t \rfloor q} + \Theta(\epsilon) \sum_{p=\lfloor t \rfloor q}^{t-1} \delta'_{t-1} + \Theta(\kappa^{-2}\epsilon) \sum_{p=\lfloor t \rfloor q}^{t-1} \mathbb{E}[\|v_{t-1}\|_2^2] + \Theta(\kappa^{-2}\epsilon).$$

Note that $\Delta'_{\lfloor t \rfloor q}$ is the estimation error of coordinate-wise estimator obtained at the beginning of each epoch, which diminishes as the batch size $S_1$ increases. Letting $S_1 = \Theta(\kappa^2/\epsilon^2)$ as specified in Theorem 2, we can bound the accumulation of $\Delta'_t$ over the all iterations as

$$\sum_{t=0}^{T-1} \Delta'_t \leq \Theta\left(\frac{1}{\kappa}\right) + \Theta(1) \sum_{t=0}^{T-1} \delta'_t + \Theta\left(\frac{1}{\kappa^2}\right) \sum_{t=0}^{T-1} \mathbb{E}[\|v_t\|_2^2] + \Theta(\kappa^{-1}). \tag{10}$$

Moreover, based on the contraction property of $\delta'_t$ provided in eq. (9), we derive the following bound for the accumulation of $\delta'_t$:

$$\sum_{t=0}^{T-1} \delta'_t \leq 2\delta'_0 + \Theta(1) \sum_{t=0}^{T-1} \Delta'_t + \Theta\left(\frac{1}{\kappa^2}\right) \sum_{t=0}^{T-1} \mathbb{E}[\|v_t\|_2^2] + \Theta(\kappa^{-4}). \tag{11}$$

Combining eq. (10) and eq. (11), the upper bounds for $\sum_{t=0}^{T-1} \Delta'_t$ and $\sum_{t=0}^{T-1} \delta'_t$ can then be derived separately as

$$\sum_{t=0}^{T-1} \Delta'_t \leq \Theta\left(\frac{1}{\kappa}\right) + \Theta(1)\delta'_0 + \Theta\left(\frac{1}{\kappa^2}\right) \sum_{t=0}^{T-1} \mathbb{E}[\|v_t\|_2^2], \tag{12}$$

$$\sum_{t=0}^{T-1} \delta'_t \leq \Theta\left(\frac{1}{\kappa}\right) + \Theta(1)\delta'_0 + \Theta\left(\frac{1}{\kappa^2}\right) \sum_{t=0}^{T-1} \mathbb{E}[\|v_t\|_2^2]. \tag{13}$$

**Step 3:** Note that eq. (12) and eq. (13) alone are not sufficient to guarantee the boundness of accumulation errors $\sum_{t=0}^{T-1} \Delta'_t$ and $\sum_{t=0}^{T-1} \delta'_t$, as the upper bounds in eq. (12) and eq. (13) depend on an unknown error term $\sum_{t=0}^{T-1} \mathbb{E}[\|v_t\|_2^2]$. To handle this issue, we utilize the Lipschitz property of $\Phi(x)$ given in Assumption 2 to obtain the following bound:

$$\left(\frac{\alpha}{2} - \frac{L\alpha^2}{2}\right) \sum_{t=0}^{T-1} \mathbb{E}[\|v_t\|_2^2] \leq \Phi(x_0) - \mathbb{E}[\Phi(x_T)] + 2\alpha\kappa^2 \sum_{t=0}^{T-1} \delta'_t + 2\alpha \sum_{t=0}^{T-1} \Delta'_t + T\Theta(\epsilon^{-2}\kappa^4). \tag{14}$$

Substituting eq. (12) and eq. (13) into eq. (14) and subtracting the residual terms on both sides yield the following bound:

$$\sum_{t=0}^{T-1} \mathbb{E}[\|v_t\|_2^2] \leq \Theta(L(\Phi(x_0) - \Phi^*)) + \Theta(\kappa). \tag{15}$$

The upper bounds of $\sum_{t=0}^{T-1} \Delta_t'$ and $\sum_{t=0}^{T-1} \delta_t'$ can then be obtained by substituting eq. (15) into eq. (12) and eq. (13).

To establish the convergence rate for $\mathbb{E}[\|\nabla\Phi(\hat{x})\|_2^2] = \frac{1}{T} \sum_{t=0}^{T} \mathbb{E}[\|\nabla\Phi(x_t)\|_2^2]$, we note that

$$\sum_{t=0}^{T-1} \mathbb{E}[\|\nabla\Phi(x_t)\|_2^2] \leq 6\kappa^2 \sum_{t=0}^{T-1} \delta_t' + 6 \sum_{t=0}^{T-1} \Delta_t' + 3 \sum_{t=0}^{T-1} \mathbb{E}[\|v_t\|_2^2] + \Theta(\kappa^{-2}). \tag{16}$$

Substituting the bounds on $\sum_{t=0}^{T-1} \mathbb{E}[\|v_t\|_2^2]$, $\sum_{t=0}^{T-1} \Delta_t'$ and $\sum_{t=0}^{T-1} \delta_t'$ into eq. (16), we obtain the convergence rate for ZO-VRGDA. □

## 5 Experiments

Our experiments focus on two types of comparisons: (a) we compare our ZO-VRGDA with other existing zeroth-order stochastic algorithms and demonstrate the superior performance of ZO-VRGDA; (b) we compare the performance of ZO-VRGDA with different inner-loop lengths.

Our experiments solve a distributionally robust optimization problem, which is commonly used for studying minimax optimization Lin et al. (2019); Rafique et al. (2018). We conduct the experiments on three datasets from LIBSVM Chang & Lin (2011). The details of the problem and the datasets are provided in Appendix A.

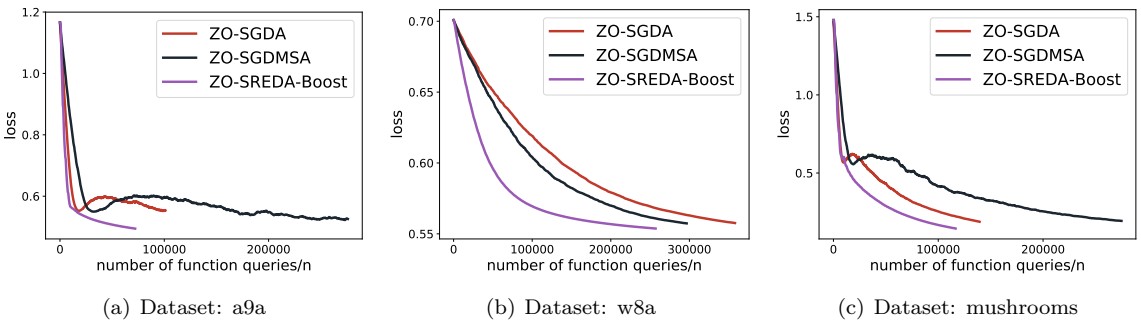

(a) Dataset: a9a      (b) Dataset: w8a      (c) Dataset: mushrooms

Figure 1: Comparison of function query complexity among three algorithms.

**Comparison among zeroth-order algorithms:** We compare the performance of our proposed ZO-VRGDA with that of two existing zeroth-order algorithms ZO-SGDA Wang et al. (2020) and ZO-SGDMSA Wang et al. (2020) designed for nonconvex-strongly-concave minimax problems. For ZO-SGDA and ZO-SGDMSA, as suggested by the corresponding theory, we set the mini-batch size $B = Cd_1/\epsilon^2$ and $B = Cd_2/\epsilon^2$ for updating the variables $x$ and $y$, respectively. For ZO-VRGDA, based on our theory, we set the mini-batch size $B = Cd_1/\epsilon$ and $B = Cd_2/\epsilon$ for updating the variables $x$ and $y$, and set $S_1 = n$ for the large batch, where $n$ is the number of data samples in the dataset. We set $C = 0.1$ and $\epsilon = 0.1$ for all algorithms. We further set the stepsize $\eta = 0.01$ for ZO-VRGDA and ZO-SGDMSA. Since ZO-SGDA is a two time-scale algorithm, we set $\eta = 0.01$ as the stepsize for the fast time scale, and $\eta/\kappa^3$ as the stepsize for slow time scale (based on the theory) where $\kappa^3 = 10$. It can be seen in Figure 1 that ZO-VRGDA substantially outperforms the other two algorithms in terms of the function query complexity (i.e., the running time).

**Comparison among different inner-loop length:** We investigate how the inner-loop length affects the overall convergence of ZO-VRGDA. We consider the following inner loop lengths $\{5, 10, 20, 50, 100\}$. It can be seen in Figure 2 that ZO-VRGDA converges faster as we increase the inner-loop length $m$ initially, and then the convergence slows down as we further enlarge $m$ beyond a certain threshold. This verifies the tradeoff role that $m$ plays, i.e., larger $m$ attains a better optimized $y$ but causes more queries. Figure 2 also illustrates that the performance of ZO-VRGDA is fairly robust to the inner-loop length as long as $m$ is not too large.

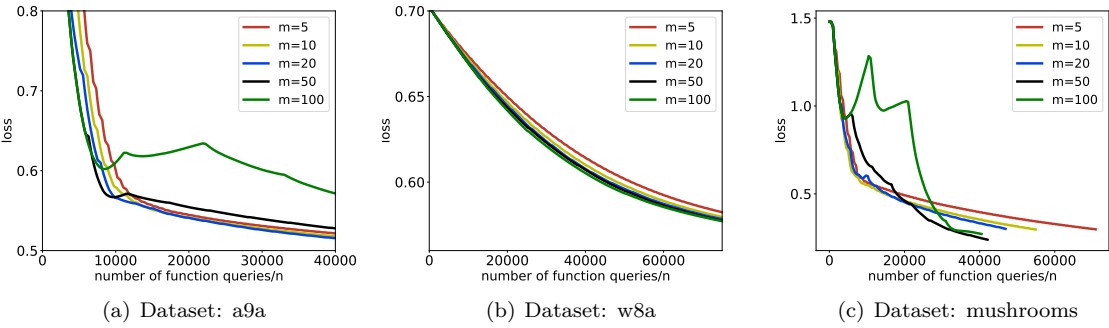

(a) Dataset: a9a        (b) Dataset: w8a        (c) Dataset: mushrooms

Figure 2: Comparison of ZO-VRGDA with different inner-loop lengths.

## 6 Conclusion

In this work, we have proposed the first zeroth-order variance reduced algorithm ZO-VRGDA for solving nonconvex-strongly-concave minimax optimization problems. The function query complexity of ZO-VRGDA achieves the best dependence on the target accuracy compared to previously designed gradient-free algorithms. We have also developed a novel analysis framework to characterize the convergence rate and the complexity, which we expect to be also useful for studying various other stochastic minimax problems such as proximal, momentum, and manifold optimization.

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

## A  Specifications of Experiments

The distributionally robust optimization problem is formulated as follows:

$$\min_{x \in \mathcal{X}} \max_{y \in \mathcal{Y}} \sum_{i=1}^{n} y_i f_i(x) - r(y),$$

where $\mathcal{X} = \{x \in \mathbb{R}^d\}$, $\mathcal{Y} = \{y \in \mathbb{R}^n | \sum_{i=1}^{n} y_i = 1, y_i \geq 0, i = 1, \cdots n\}$, $r(y) = 10 \sum_{i=1}^{n} (y_i - 1/n)^2$, $f_i(x) = \phi(l(x))$ where $\phi(\theta) = 2\log\left(1 + \frac{\theta}{2}\right)$, $l(x; s, z) = \log(1 + \exp(-zx^\top s))$, and $(s, z)$ are the feature and label pair of a data sample. It can be seen that the problem is a minimax problem with $d_1 = d$ and $d_2 = n$. Since the distributionally robust optimization aims at an unbalanced dataset, we pick the samples from the original dataset and set the ratio between the number of negative labeled samples and the number of positive labeled samples to be $1 : 4$. Since the maximization over $y$ is a constrained optimization problem, we incorporate a projection step after updates of $y$ for all algorithms.

The details of the datasets used for zeroth-order algorithms are listed in Table 2.

Table 2: Datasets used for zeroth-order algorithms

| Datasets | # of samples | # of features | # Pos: # Neg |
|---|---|---|---|
| mushrooms | 200 | 112 | 1:4 |
| w8a | 100 | 300 | 1:4 |
| a9a | 150 | 123 | 1:4 |

## B  Technical Lemmas

### B.1  Preliminary Lemmas

We first provide useful inequalities in convex optimization Nesterov (2013); Polyak (1963) and auxiliary lemmas from Fang et al. (2018); Luo et al. (2020).

**Lemma 2** (Nesterov (2013),Polyak (1963))**.** *Suppose $h(\cdot)$ is convex and has $\ell$-Lipschitz gradient. Then, we have*

$$\langle \nabla h(w) - \nabla h(w'), w - w' \rangle \geq \frac{1}{\ell} \|\nabla h(w) - \nabla h(w')\|_2^2. \tag{17}$$

**Lemma 3** (Nesterov (2013),Polyak (1963))**.** *Suppose $h(\cdot)$ is $\mu$-strongly convex and has $\ell$-Lipschitz gradient. Let $w^*$ be the minimizer of $h$. Then for any $w$ and $w'$, the following inequalities hold:*

$$\langle \nabla h(w) - \nabla h(w'), w - w' \rangle \geq \frac{\mu\ell}{\mu + \ell} \|w - w'\|_2^2 + \frac{1}{\mu + \ell} \|\nabla h(w) - \nabla h(w')\|_2^2, \tag{18}$$

$$\|\nabla h(w) - \nabla h(w')\|_2 \geq \mu \|w - w'\|_2, \tag{19}$$

$$2\mu(h(w) - h(w')) \leq \|\nabla h(w)\|_2^2. \tag{20}$$

**Lemma 4** (Fang et al. (2018), Lemma 2)**.** *Suppose Assumption 4 hold. For any $(x, y) \in \mathbb{R}^{d_1} \times \mathbb{R}^{d_2}$ and sample batch $\{\xi_1, \cdots, \xi_S\}$, let $v = \frac{1}{S} \sum_{i=1}^{S} \nabla_x F(x, y, \xi_i)$ and $u = \frac{1}{S} \sum_{i=1}^{S} \nabla_y F(x, y, \xi_i)$. We have*

$$\mathbb{E}[\|v - \nabla_x f(x, y)\|_2^2] + \mathbb{E}[\|u - \nabla_y f(x, y)\|_2^2] \leq \frac{\sigma^2}{S}.$$

**Lemma 5** (Fang et al. (2018), Lemma 1)**.** *Let $\mathcal{V}_t$ be an estimator of $\mathcal{B}(z_t)$ as*

$$\mathcal{V}_t = \mathcal{B}_{\mathcal{S}_*}(z_t) - \mathcal{B}_{\mathcal{S}_*}(z_{t-1}) + \mathcal{V}_{t-1},$$

*where $\mathcal{B}_{\mathcal{S}_*} = \frac{1}{|\mathcal{S}_*|}\sum_{\mathcal{B}_i \in \mathcal{S}_*} \mathcal{B}_i$ satisfies*

$$\mathbb{E}[\mathcal{B}_i(z_t) - \mathcal{B}_i(z_{t-1})|z_0, \cdots, z_{t-1}] = \mathbb{E}[\mathcal{V}_t - \mathcal{V}_{t-1}|z_0, \cdots, z_{t-1}].$$

*For all $k = 1, \cdots, K$, we have*

$$\mathbb{E}[\|\mathcal{V}_t - \mathcal{V}_{t-1} - (\mathcal{B}_{\mathcal{S}_*}(z_t) - \mathcal{B}_{\mathcal{S}_*}(z_{t-1}))\|_2^2] \leq \frac{1}{\mathcal{S}_*}\mathbb{E}[\|\mathcal{B}_i(z_t) - \mathcal{B}_i(z_{t-1})\|_2^2 |z_0, \cdots, z_{t-1}],$$

*and*

$$\mathbb{E}[\|\mathcal{V}_t - \mathcal{B}(z_t)|z_0, \cdots, z_{t-1}\|_2^2] \leq \|\mathcal{V}_{t-1} - \mathcal{B}(z_{t-1})\|_2^2 + \frac{1}{|\mathcal{S}_*|}\mathbb{E}[\|\mathcal{B}_i(z_t) - \mathcal{B}_i(z_{t-1})\|_2^2 |z_0, \cdots, z_{t-1}].$$

*Furthermore, if $\mathcal{B}_i$ is L-Lipschitz continuous in expectation, we have*

$$\mathbb{E}[\|\mathcal{V}_t - \mathcal{B}(z_t)|z_0, \cdots, z_{t-1}\|_2^2] \leq \|\mathcal{V}_{t-1} - \mathcal{B}(z_{t-1})\|_2^2 + \frac{L^2}{|\mathcal{S}_*|}\mathbb{E}[\|z_t - z_{t-1}\|_2^2 |z_0, \cdots, z_{t-1}].$$

We provide the following lemmas to characterize the properties of Gaussian smoothed function and zeroth-order Gaussian gradient estimator. Consider a function $h(\cdot): \mathbb{R}^d \to \mathbb{R}$. Let $\nu$ be a $d$-dimensional standard Gaussian random vector and $\mu > 0$ be the smoothing parameter. Then a smooth approximation of $h(\cdot)$ is defined as $h_\tau(x) = \mathbb{E}_\nu[h(x + \tau\nu)]$. We have the following lemmas.

**Lemma 6** (Nesterov & Spokoiny (2017), Section 2)**.** *If $h(\cdot)$ is convex, then $h_\mu(\cdot)$ is also a convex function.*

**Lemma 7** (Ghadimi & Lan (2013), Section 3.1)**.** *If $h(\cdot)$ has $\ell$-Lipschitz gradient, then $h_\mu(\cdot)$ also has $\ell$-Lipschitz gradient.*

**Lemma 8** (Nesterov & Spokoiny (2017), Theorem 1)**.** *If $h(\cdot)$ has $\ell$-Lipschitz gradient, then for all $x \in \mathbb{R}^d$, we have $|h(x) - h_\tau(x)| \leq \frac{\tau^2}{2}\ell d$.*

**Lemma 9** (Nesterov & Spokoiny (2017), Lemma 3)**.** *If $h(\cdot)$ has $\ell$-Lipschitz gradient, then $\|\nabla_x h_\tau(x) - \nabla_x h(x)\|_2^2 \leq \frac{\tau^2}{4}\ell^2(d+3)^3$.*

The following lemma characterizes the estimation error of a zeroth-order coordinate-wise estimator with batch size $S_1$ in lines 6 and 7 in Algorithm 1.

**Lemma 10.** *Suppose Assumption 2 and 4 hold. Suppose $mod(t, q) = 0$, and let $\epsilon(S_1, \delta) = \mathbb{E}[\|v_t - \nabla_x f_{\mu_1}(x_t, y_t)\|_2^2] + \mathbb{E}[\|u_t - \nabla_y f_{\mu_2}(x_t, y_t)\|_2^2]$. Then, we have*

$$\epsilon(S_1, \delta) \leq \frac{(d_1 + d_2)\ell^2\delta^2}{2} + \frac{4\sigma^2}{S_1} + \frac{\mu_1^2}{2}\ell^2(d_1+3)^3 + \frac{\mu_2^2}{2}\ell^2(d_2+3)^3.$$

*Proof.* (B.56) and (B.57) in Fang et al. (2018) imply that

$$\mathbb{E}[\|v_t - \nabla_x f(x_t, y_t)\|_2^2] \leq \frac{d_1\ell^2\delta^2}{2} + \frac{2\sigma^2}{S_1}, \tag{21}$$

and

$$\mathbb{E}[\|u_t - \nabla_y f(x_t, y_t)\|_2^2] \leq \frac{d_2\ell^2\delta^2}{2} + \frac{2\sigma^2}{S_1}. \tag{22}$$

Then we proceed as follows:

$$\mathbb{E}[\|v_t - \nabla_x f_{\mu_1}(x_t, y_t)\|_2^2] + \mathbb{E}[\|u_t - \nabla_y f_{\mu_2}(x_t, y_t)\|_2^2]$$
$$\leq 2\mathbb{E}[\|v_t - \nabla_x f(x_t, y_t)\|_2^2] + 2\mathbb{E}[\|u_t - \nabla_y f(x_t, y_t)\|_2^2]$$
$$+ 2\mathbb{E}[\|\nabla_x f_{\mu_1}(x_t, y_t) - \nabla_x f(x_t, y_t)\|_2^2] + 2\mathbb{E}[\|\nabla_x f_{\mu_2}(x_t, y_t) - \nabla_y f(x_t, y_t)\|_2^2]$$

$$\overset{(i)}{\leq} 2\mathbb{E}[\|v_t - \nabla_x f(x_t, y_t)\|_2^2] + 2\mathbb{E}[\|u_t - \nabla_y f(x_t, y_t)\|_2^2] + \frac{\mu_1^2}{2}\ell^2(d_1+3)^3 + \frac{\mu_2^2}{2}\ell^2(d_2+3)^3$$

$$\overset{(ii)}{\leq} (d_1+d_2)\ell^2\delta^2 + \frac{8\sigma^2}{S_1} + \frac{\mu_1^2}{2}\ell^2(d_1+3)^3 + \frac{\mu_2^2}{2}\ell^2(d_2+3)^3,$$

where $(i)$ follows from Lemma 9, and $(ii)$ follows from eq. (21) and eq. (22). $\qquad\square$

We denote

$$G_{\mu_1}(x, y, \nu_i, \xi_i) = \frac{F(x+\mu_1\nu_i, y, \xi_i) - F(x, y, \xi_i)}{\mu_1}\nu_i$$

and

$$H_{\mu_2}(x, y, \omega_i, \xi_i) = \frac{F(x, y+\mu_2\omega_i, \xi_i) - F(x, y, \xi_i)}{\mu_2}\omega_i$$

as unbiased estimators of $\nabla_x f_{\mu_1}(x, y)$ and $\nabla_y f_{\mu_2}(x, y)$, respectively. Then we have the following lemma.

**Lemma 11.** *Suppose Assumption 2 holds, and suppose $u_1$ and $u_2$ are standard Gaussian random vector, i.e., $\nu_i \sim N(0, \mathbf{1}_{d_1})$ and $\omega_i \sim N(0, \mathbf{1}_{d_2})$. Then, we have*

$$\mathbb{E}\left[\|G_{\mu_1}(x, y, \nu_i, \xi_i) - G_{\mu_1}(x', y, \nu_i, \xi_i)\|_2^2\right] \leq 2(d_1+4)\ell^2\|x-x'\|_2^2 + 2\mu_1^2(d_1+6)^3\ell^2,$$

$$\mathbb{E}\left[\|G_{\mu_1}(x, y, \nu_i, \xi_i) - G_{\mu_1}(x, y', \nu_i, \xi_i)\|_2^2\right] \leq 2(d_1+4)\ell^2\|y-y'\|_2^2 + 2\mu_1^2(d_1+6)^3\ell^2,$$

*and*

$$\mathbb{E}\left[\|H_{\mu_2}(x, y, \nu_i, \xi_i) - H_{\mu_2}(x', y, \nu_i, \xi_i)\|_2^2\right] \leq 2(d_2+4)\ell^2\|x-x'\|_2^2 + 2\mu_2^2(d_2+6)^3\ell^2,$$

$$\mathbb{E}\left[\|H_{\mu_2}(x, y, \nu_i, \xi_i) - H_{\mu_2}(x, y', \nu_i, \xi_i)\|_2^2\right] \leq 2(d_2+4)\ell^2\|y-y'\|_2^2 + 2\mu_2^2(d_2+6)^3\ell^2.$$

*Proof.* The proof is similar to that of Lemma 3 in Fang et al. (2018). Here we provide the proof for completeness. We will show how to upper bound the term $\mathbb{E}\left[\|G_{\mu_1}(x, y, \nu_1, \xi) - G_{\mu_1}(x', y, \nu_1, \xi)\|_2^2\right]$ here. Then, the upper bounds on the remaining three terms can be obtained by following similar steps. We proceed the bound as follows.

$$\mathbb{E}\left[\|G_{\mu_1}(x, y, \nu_i, \xi_i) - G_{\mu_1}(x, y', \nu_i, \xi_i)\|_2^2\right]$$

$$= \mathbb{E}\left[\left\|\frac{F(x+\mu_1\nu_i, y, \xi_i) - F(x, y, \xi_i)}{\mu_1}\nu_1 - \frac{F(x+\mu_1\nu_i, y', \xi_i) - F(x, y', \xi_i)}{\mu_1}\nu_1\right\|_2^2\right]$$

$$= \mathbb{E}\left[\left\|\frac{F(x+\mu_1\nu_i, y, \xi_i) - F(x, y, \xi_i) - \langle\nabla_x F(x, y, \xi_i), \mu_1\nu_i\rangle}{\mu_1}\nu_i\right.\right.$$

$$- \frac{F(x+\mu_1\nu_i, y', \xi_i) - F(x, y', \xi_i) - \langle\nabla_x F(x, y', \xi_i), \mu_1\nu_i\rangle}{\mu_1}\nu_i$$

$$\left.\left.+ \langle\nabla_x F(x, y, \xi_i) - \nabla_x F(x, y', \xi_i), \nu_i\rangle\nu_i\right\|_2^2\right]$$

$$\leq 2\mathbb{E}\left[\left\|\frac{F(x+\mu_1\nu_i, y, \xi_i) - F(x, y, \xi_i) - \langle\nabla_x F(x, y, \xi_i), \mu_1\nu_i\rangle}{\mu_1}\nu_i\right.\right.$$

$$\left.\left.- \frac{F(x+\mu_1\nu_i, y', \xi_i) - F(x, y', \xi_i) - \langle\nabla_x F(x, y', \xi_i), \mu_1\nu_i\rangle}{\mu_1}\nu_i\right\|_2^2\right]$$

$$+ 2\mathbb{E}\left[\|\langle\nabla_x F(x, y, \xi_i) - \nabla_x F(x, y', \xi_i), \nu_i\rangle\nu_i\|_2^2\right]$$

$$\leq 4\mathbb{E}\left[\left\|\frac{F(x+\mu_1\nu_i,y,\xi_i)-F(x,y,\xi_i)-\langle\nabla_xF(x,y,\xi_i),\mu_1\nu_i\rangle}{\mu_1}\nu_i\right\|_2^2\right]$$

$$+4\mathbb{E}\left[\left\|\frac{F(x+\mu_1\nu_i,y',\xi_i)-F(x,y',\xi_i)-\langle\nabla_xF(x,y',\xi_i),\mu_1\nu_i\rangle}{\mu_1}\nu_i\right\|_2^2\right]$$

$$+2\mathbb{E}\left[\|\langle\nabla_xF(x,y,\xi_i)-\nabla_xF(x,y',\xi_i),\nu_i\rangle\nu_i\|_2^2\right]$$

$$\leq 4\mathbb{E}\left[\left|\frac{F(x+\mu_1\nu_i,y,\xi_i)-F(x,y,\xi_i)-\langle\nabla_xF(x,y,\xi_i),\mu_1\nu_i\rangle}{\mu_1}\right|^2\|\nu_i\|_2^2\right]$$

$$+4\mathbb{E}\left[\left|\frac{F(x+\mu_1\nu_i,y',\xi_i)-F(x,y',\xi_i)-\langle\nabla_xF(x,y',\xi_i),\mu_1\nu_i\rangle}{\mu_1}\right|^2\|\nu_i\|_2^2\right]$$

$$+2\mathbb{E}\left[\|\langle\nabla_xF(x,y,\xi_i)-\nabla_xF(x,y',\xi_i),\nu_i\rangle\nu_i\|_2^2\right]$$

$$\overset{(i)}{\leq}2\mu_1^2\ell^2\mathbb{E}[\|\nu_i\|_2^2]+2\mathbb{E}\left[\|\langle\nabla_xF(x,y,\xi_i)-\nabla_xF(x,y',\xi_i),\nu_i\rangle\nu_i\|_2^2\right]$$

$$\overset{(ii)}{\leq}2\mu_1^2\ell^2\mathbb{E}[\|\nu_i\|_2^2]+2(d_1+4)\mathbb{E}\left[\|\nabla_xF(x,y,\xi_i)-\nabla_xF(x,y',\xi_i)\|_2^2\right]$$

$$\overset{(iii)}{\leq}2\mu_1^2(d_1+6)^3\ell^2+2(d_1+4)\ell^2\mathbb{E}\left[\|y-y'\|_2^2\right],$$

where $(i)$ follows from the fact that for any $a,a'\in\mathbb{R}^{d_1}$ and $b\in\mathbb{R}^{d_2}$, we have

$$|F(a,b,\xi_i)-F(a',b,\xi_i)-\langle\nabla_xF(a,b,\xi_i),a-a'\rangle|\leq\frac{\ell}{2}\|a-a'\|_2^2,$$

because $F(a,b,\xi)$ has $\ell$-Lipschitz continuous gradient; $(ii)$ follows because

$$\mathbb{E}[\|\langle a,\nu_i\rangle\nu_i\|_2^2]\leq(d_1+4)\|a\|_2^2,$$

obtained from (33) in Nesterov & Spokoiny (2017), and $(iii)$ follows because $\mathbb{E}[\|\nu_i\|_2^2]\leq(d_1+6)^3$ in (17) of Nesterov & Spokoiny (2017). □

## B.2 Useful Properties for Zeroth-Order Concave Maximizer

In this section, we show some properties for the zeroth-order concave maximizer in Algorithm 3. For simplicity, for any given $t\geq0$, we define $g_t(y)=-f(x_{t+1},y)$ and $g_{t,\mu_2}(y)=-f_{\mu_2}(x_{t+1},y)$. Lemma 6 and Lemma 7 imply that $g_t(\cdot)$ is $\mu$-strongly convex and has $\ell$-Lipschitz gradient, and $g_{t,\mu_2}(\cdot)$ is convex and has $\ell$-Lipschitz gradient. We also define $\tilde{y}_t^*=\arg\min_y g_t(y)$. We can obtain the following two lemmas by following the same steps in Luo et al. (2020)

**Lemma 12** (Lemma 9 of Luo et al. (2020)). *Consider Algorithm 3. We have*

$$\sum_{k=0}^m\mathbb{E}[\|\nabla g_{t,\mu_2}(\tilde{y}_{t,k})\|_2^2]\leq\frac{2}{\beta}\mathbb{E}[g_{t,\mu_2}(\tilde{y}_{t,0})-g_{t,\mu_2}(\tilde{y}_{t,m+1})]+\sum_{k=0}^m\mathbb{E}[\|\nabla g_{t,\mu_2}(\tilde{y}_{t,k})-\tilde{u}_{t,k}\|_2^2].$$

**Lemma 13** (Lemma 11 of Luo et al. (2020)). *Consider Algorithm 3 with any $\beta\leq\frac{2}{\ell}$ and $k\geq1$. We have*

$$\mathbb{E}[\|\nabla g_{t,\mu_2}(\tilde{y}_{t,k})-\tilde{u}_{t,k}\|_2^2]\leq\mathbb{E}[\|\nabla g_{t,\mu_2}(\tilde{y}_{t,0})-\tilde{u}_{t,0}\|_2^2]+\frac{\ell\beta}{2-\ell\beta}\mathbb{E}[\|\tilde{u}_{t,0}\|_2^2].$$

The following lemma characterizes the recursion of $\mathbb{E}[\|\nabla g_{t,\mu_2}(\tilde{y}_{t,\tilde{m}_t})\|_2^2]$ within each inner loop.

**Lemma 14.** *Consider Algorithm 3. For any $k\geq1$ and $\beta\leq\frac{1}{\ell}$, we have*

$$\mathbb{E}[\|\nabla g_{t,\mu_2}(\tilde{y}_{t,\tilde{m}_t})\|_2^2]\leq\frac{2}{\beta\mu(m+1)}\mathbb{E}[\|\nabla g_{t,\mu_2}(\tilde{y}_{t,0})\|_2^2]+\mathbb{E}[\|\nabla g_{t,\mu_2}(\tilde{y}_{t,0})-\tilde{u}_{t,0}\|_2^2]+\frac{\ell\beta}{2-\ell\beta}\mathbb{E}[\|\tilde{u}_{t,0}\|_2^2]$$

$$+\frac{2}{\beta(m+1)}\left(\frac{\mu_2^2}{4\mu}\ell^2(d_2+3)^3+\mu_2^2\ell d_2\right).$$

*Proof.* Taking summation of the result of Lemma 13 over $t = \{0, \cdots m\}$ yields

$$\sum_{k=0}^{m} \mathbb{E}[\|\nabla g_{t,\mu_2}(\tilde{y}_{t,k}) - \tilde{u}_{t,k}\|_2^2] \leq (m+1)\mathbb{E}[\|\nabla g_{t,\mu_2}(\tilde{y}_{t,0}) - \tilde{u}_{t,0}\|_2^2] + \frac{\ell\beta(m+1)}{2 - \ell\beta}\mathbb{E}[\|\tilde{u}_{t,0}\|_2^2]. \tag{23}$$

Combining eq. (23) with Lemma 12 yields

$$\sum_{k=0}^{m} \mathbb{E}[\|\nabla g_{t,\mu_2}(\tilde{y}_{t,k})\|_2^2] \leq \frac{2}{\beta}\mathbb{E}[g_{t,\mu_2}(\tilde{y}_{t,0}) - g_{t,\mu_2}(\tilde{y}_{t,m+1})] + (m+1)\mathbb{E}[\|\nabla g_{t,\mu_2}(\tilde{y}_{t,0}) - \tilde{u}_{t,0}\|_2^2]$$
$$+ \frac{\ell\beta(m+1)}{2 - \ell\beta}\mathbb{E}[\|\tilde{u}_{t,0}\|_2^2]. \tag{24}$$

Dividing both sides of eq. (24) by $m$ and recalling the definition of $\tilde{m}_t$ in the output of Algorithm 3 yield

$$\mathbb{E}[\|\nabla g_{t,\mu_2}(\tilde{y}_{t,\tilde{m}_t})\|_2^2] \leq \frac{2}{\beta(m+1)}\mathbb{E}[g_{t,\mu_2}(\tilde{y}_{t,0}) - g_{t,\mu_2}(\tilde{y}_{t,m+1})] + \mathbb{E}[\|\nabla g_{t,\mu_2}(\tilde{y}_{t,0}) - \tilde{u}_{t,0}\|_2^2]$$
$$+ \frac{\ell\beta}{2 - \ell\beta}\mathbb{E}[\|\tilde{u}_{t,0}\|_2^2]. \tag{25}$$

We then bound the term $\mathbb{E}[g_{t,\mu_2}(\tilde{y}_{t,0}) - g_{t,\mu_2}(\tilde{y}_{t,m+1})]$ as follows:

$$\mathbb{E}[g_{t,\mu_2}(\tilde{y}_{t,0}) - g_{t,\mu_2}(\tilde{y}_{t,m+1})]$$
$$= \mathbb{E}[g_t(\tilde{y}_{t,0}) - g_t(\tilde{y}_{t,m+1})] + \mathbb{E}[g_{t,\mu_2}(\tilde{y}_{t,0}) - g_t(\tilde{y}_{t,0})] + \mathbb{E}[g_t(\tilde{y}_{t,m+1}) - g_{t,\mu_2}(\tilde{y}_{t,m+1})]$$
$$\leq \mathbb{E}[g_t(\tilde{y}_{t,0}) - g_t(\tilde{y}_{t,m+1})] + \mathbb{E}[|g_{t,\mu_2}(\tilde{y}_{t,0}) - g_t(\tilde{y}_{t,0})|] + \mathbb{E}[|g_{t,\mu_2}(\tilde{y}_{t,m+1}) - g_t(\tilde{y}_{t,m+1})|]$$
$$\overset{(i)}{\leq} \mathbb{E}[g_t(\tilde{y}_{t,0}) - g_t(\tilde{y}_{t,m+1})] + \mu_2^2\ell d_2$$
$$\leq \mathbb{E}[g_t(\tilde{y}_{t,0}) - g_t(\tilde{y}_t^*)] + \mu_2^2\ell d_2$$
$$\overset{(ii)}{\leq} \frac{1}{2\mu}\mathbb{E}[\|\nabla g_t(\tilde{y}_{t,0})\|_2^2] + \mu_2^2\ell d_2$$
$$\leq \frac{1}{\mu}\mathbb{E}[\|\nabla g_{t,\mu_2}(\tilde{y}_{t,0})\|_2^2] + \frac{1}{\mu}\mathbb{E}[\|\nabla g_{t,\mu_2}(\tilde{y}_{t,0}) - \nabla g_t(\tilde{y}_{t,0})\|_2^2] + \mu_2^2\ell d_2$$
$$\overset{(iii)}{\leq} \frac{1}{\mu}\mathbb{E}[\|\nabla g_{t,\mu_2}(\tilde{y}_{t,0})\|_2^2] + \frac{\mu_2^2}{4\mu}\ell^2(d_2 + 3)^3 + \mu_2^2\ell d_2, \tag{26}$$

where $(i)$ follows from Lemma 8, $(ii)$ follows from eq. (20) in Lemma 3, and $(iii)$ follows from Lemma 9. Substituting eq. (26) into eq. (25) yields

$$\mathbb{E}[\|\nabla g_{t,\mu_2}(\tilde{y}_{t,\tilde{m}_t})\|_2^2] \leq \frac{2}{\beta\mu(m+1)}\mathbb{E}[\|\nabla g_{t,\mu_2}(\tilde{y}_{t,0})\|_2^2] + \mathbb{E}[\|\nabla g_{t,\mu_2}(\tilde{y}_{t,0}) - \tilde{u}_{t,0}\|_2^2] + \frac{\ell\beta}{2 - \ell\beta}\mathbb{E}[\|\tilde{u}_{t,0}\|_2^2]$$
$$+ \frac{2}{\beta(m+1)}\left(\frac{\mu_2^2}{4\mu}\ell^2(d_2 + 3)^3 + \mu_2^2\ell d_2\right),$$

which completes the proof. $\qquad\square$

**Lemma 15.** *Consider Algorithm 3. Let $S_{2,y} \geq 16\kappa(d_2 + 4)\ell\beta$ and $\beta \leq \frac{1}{6\ell}$. For any $t > 0$, we have*

$$\sum_{k=0}^{m} \mathbb{E}[\|\tilde{u}_{t,k}\|_2^2] \leq \frac{1}{1-b}\mathbb{E}[\|\tilde{u}_{t,0}\|_2^2] + \frac{m+1}{1-b}\left[\frac{2\mu_2^2\ell\kappa}{\beta}(d_2 + 3)^3 + 7\mu_2^2(d_2 + 6)^3\ell^2\right],$$

*where $b = 1 - \frac{\beta\mu\ell}{2(\mu+\ell)}$.*

*Proof.* The update of Algorithm 3 implies that

$$\mathbb{E}[\|\tilde{u}_{t,k}\|_2^2 | \mathcal{F}_{t,k}]$$

$$= \|\tilde{u}_{t,k-1}\|_2^2 + 2\mathbb{E}[\langle \tilde{u}_{t,k-1}, H_{\mu_2}(\tilde{x}_{t,k}, \tilde{y}_{t,k}, \omega_{\mathcal{M}_2}, \xi_{\mathcal{M}}) - H_{\mu_2}(\tilde{x}_{t,k-1}, \tilde{y}_{t,k-1}, \omega_{\mathcal{M}_2}, \xi_{\mathcal{M}})\rangle | \mathcal{F}_{t,k}]$$
$$+ \mathbb{E}[\|H_{\mu_2}(\tilde{x}_{t,k}, \tilde{y}_{t,k}, \omega_{\mathcal{M}_2}, \xi_{\mathcal{M}}) - H_{\mu_2}(\tilde{x}_{t,k-1}, \tilde{y}_{t,k-1}, \omega_{\mathcal{M}_2}, \xi_{\mathcal{M}})\|_2^2 | \mathcal{F}_{t,k}]$$

$$= \|\tilde{u}_{t,k-1}\|_2^2 + \frac{2}{\beta}\langle \tilde{y}_{t,k} - \tilde{y}_{t,k-1}, \nabla_y f_{\mu_2}(\tilde{x}_{t,k}, \tilde{y}_{t,k}) - \nabla_y f_{\mu_2}(\tilde{x}_{t,k-1}, \tilde{y}_{t,k-1})\rangle$$
$$+ \mathbb{E}[\|H_{\mu_2}(\tilde{x}_{t,k}, \tilde{y}_{t,k}, \omega_{\mathcal{M}_2}, \xi_{\mathcal{M}}) - H_{\mu_2}(\tilde{x}_{t,k-1}, \tilde{y}_{t,k-1}, \omega_{\mathcal{M}_2}, \xi_{\mathcal{M}})\|_2^2 | \mathcal{F}_{t,k}]$$

$$= \|\tilde{u}_{t,k-1}\|_2^2 + \frac{2}{\beta}\langle \tilde{y}_{t,k} - \tilde{y}_{t,k-1}, \nabla_y f(\tilde{x}_{t,k}, \tilde{y}_{t,k}) - \nabla_y f(\tilde{x}_{t,k-1}, \tilde{y}_{t,k-1})\rangle$$
$$+ \frac{2}{\beta}\langle \tilde{y}_{t,k} - \tilde{y}_{t,k-1}, \nabla_y f_{\mu_2}(\tilde{x}_{t,k}, \tilde{y}_{t,k}) - \nabla_y f(\tilde{x}_{t,k}, \tilde{y}_{t,k})\rangle$$
$$+ \frac{2}{\beta}\langle \tilde{y}_{t,k} - \tilde{y}_{t,k-1}, \nabla_y f(\tilde{x}_{t,k-1}, \tilde{y}_{t,k-1}) - \nabla_y f_{\mu_2}(\tilde{x}_{t,k-1}, \tilde{y}_{t,k-1})\rangle$$
$$+ \mathbb{E}[\|H_{\mu_2}(\tilde{x}_{t,k}, \tilde{y}_{t,k}, \omega_{\mathcal{M}_2}, \xi_{\mathcal{M}}) - H_{\mu_2}(\tilde{x}_{t,k-1}, \tilde{y}_{t,k-1}, \omega_{\mathcal{M}_2}, \xi_{\mathcal{M}})\|_2^2 | \mathcal{F}_{t,k}]$$

$$\overset{(i)}{\leq} \|\tilde{u}_{t,k-1}\|_2^2 - \frac{2}{\beta}\left(\frac{\mu\ell}{\mu+\ell}\|\tilde{y}_{t,k} - \tilde{y}_{t,k-1}\|_2^2 + \frac{1}{\mu+\ell}\|\nabla_y f(\tilde{x}_{t,k}, \tilde{y}_{t,k}) - \nabla_y f(\tilde{x}_{t,k-1}, \tilde{y}_{t,k-1})\|_2^2\right)$$
$$+ \frac{2}{\beta}\left(\frac{\mu\ell}{4(\mu+\ell)}\|\tilde{y}_{t,k} - \tilde{y}_{t,k-1}\|_2^2 + \frac{\mu+\ell}{\mu\ell}\|\nabla_y f_{\mu_2}(\tilde{x}_{t,k}, \tilde{y}_{t,k}) - \nabla_y f(\tilde{x}_{t,k}, \tilde{y}_{t,k})\|_2^2\right)$$
$$+ \frac{2}{\beta}\left(\frac{\mu\ell}{4(\mu+\ell)}\|\tilde{y}_{t,k} - \tilde{y}_{t,k-1}\|_2^2 + \frac{\mu+\ell}{\mu\ell}\|\nabla_y f_{\mu_2}(\tilde{x}_{t,k-1}, \tilde{y}_{t,k-1}) - \nabla_y f(\tilde{x}_{t,k-1}, \tilde{y}_{t,k-1})\|_2^2\right)$$
$$+ \mathbb{E}[\|H_{\mu_2}(\tilde{x}_{t,k}, \tilde{y}_{t,k}, \omega_{\mathcal{M}_2}, \xi_{\mathcal{M}}) - H_{\mu_2}(\tilde{x}_{t,k-1}, \tilde{y}_{t,k-1}, \omega_{\mathcal{M}_2}, \xi_{\mathcal{M}})\|_2^2 | \mathcal{F}_{t,k}]$$

$$\overset{(ii)}{\leq} \|\tilde{u}_{t,k-1}\|_2^2 - \frac{\mu\ell}{\beta(\mu+\ell)}\|\tilde{y}_{t,k} - \tilde{y}_{t,k-1}\|_2^2 - \frac{2}{\beta(\mu+\ell)}\|\nabla_y f(\tilde{x}_{t,k}, \tilde{y}_{t,k}) - \nabla_y f(\tilde{x}_{t,k-1}, \tilde{y}_{t,k-1})\|_2^2$$
$$+ \frac{\mu_2^2 \ell(\mu+\ell)}{\beta\mu}(d_2+3)^3 + \mathbb{E}[\|H_{\mu_2}(\tilde{x}_{t,k}, \tilde{y}_{t,k}, \omega_{\mathcal{M}_2}, \xi_{\mathcal{M}}) - H_{\mu_2}(\tilde{x}_{t,k-1}, \tilde{y}_{t,k-1}, \omega_{\mathcal{M}_2}, \xi_{\mathcal{M}})\|_2^2 | \mathcal{F}_{t,k}]$$

$$\leq \left(1 - \frac{\beta\mu\ell}{\mu+\ell}\right)\|\tilde{u}_{t,k-1}\|_2^2 - \frac{2}{\beta(\mu+\ell)}\|\nabla_y f(\tilde{x}_{t,k}, \tilde{y}_{t,k}) - \nabla_y f(\tilde{x}_{t,k-1}, \tilde{y}_{t,k-1})\|_2^2$$
$$+ 2\mathbb{E}[\|H_{\mu_2}(\tilde{x}_{t,k}, \tilde{y}_{t,k}, \omega_{\mathcal{M}_2}, \xi_{\mathcal{M}}) - H_{\mu_2}(\tilde{x}_{t,k-1}, \tilde{y}_{t,k-1}, \omega_{\mathcal{M}_2}, \xi_{\mathcal{M}})$$
$$- (\nabla_y f_{\mu_2}(\tilde{x}_{t,k}, \tilde{y}_{t,k}) - \nabla_y f_{\mu_2}(\tilde{x}_{t,k-1}, \tilde{y}_{t,k-1}))\|_2^2 | \mathcal{F}_{t,k}]$$
$$+ 2\mathbb{E}[\|\nabla_y f_{\mu_2}(\tilde{x}_{t,k}, \tilde{y}_{t,k}) - \nabla_y f_{\mu_2}(\tilde{x}_{t,k-1}, \tilde{y}_{t,k-1})\|_2^2 | \mathcal{F}_{t,k}] + \frac{\mu_2^2 \ell(\mu+\ell)}{\beta\mu}(d_2+3)^3$$

$$\leq \left(1 - \frac{\beta\mu\ell}{\mu+\ell}\right)\|\tilde{u}_{t,k-1}\|_2^2 - \frac{2}{\beta(\mu+\ell)}\|\nabla_y f(\tilde{x}_{t,k}, \tilde{y}_{t,k}) - \nabla_y f(\tilde{x}_{t,k-1}, \tilde{y}_{t,k-1})\|_2^2$$
$$+ 2\mathbb{E}[\|H_{\mu_2}(\tilde{x}_{t,k}, \tilde{y}_{t,k}, \omega_{\mathcal{M}_2}, \xi_{\mathcal{M}}) - H_{\mu_2}(\tilde{x}_{t,k-1}, \tilde{y}_{t,k-1}, \omega_{\mathcal{M}_2}, \xi_{\mathcal{M}})$$
$$- (\nabla_y f_{\mu_2}(\tilde{x}_{t,k}, \tilde{y}_{t,k}) - \nabla_y f_{\mu_2}(\tilde{x}_{t,k-1}, \tilde{y}_{t,k-1}))\|_2^2 | \mathcal{F}_{t,k}]$$
$$+ 6\mathbb{E}[\|\nabla_y f(\tilde{x}_{t,k}, \tilde{y}_{t,k}) - \nabla_y f(\tilde{x}_{t,k-1}, \tilde{y}_{t,k-1})\|_2^2 | \mathcal{F}_{t,k}]$$
$$+ 6\mathbb{E}[\|\nabla_y f(\tilde{x}_{t,k-1}, \tilde{y}_{t,k-1}) - \nabla_y f_{\mu_2}(\tilde{x}_{t,k-1}, \tilde{y}_{t,k-1})\|_2^2 | \mathcal{F}_{t,k}]$$
$$+ 6\mathbb{E}[\|\nabla_y f_{\mu_2}(\tilde{x}_{t,k}, \tilde{y}_{t,k}) - \nabla_y f(\tilde{x}_{t,k}, \tilde{y}_{t,k})\|_2^2 | \mathcal{F}_{t,k}] + \frac{\mu_2^2 \ell(\mu+\ell)}{\beta\mu}(d_2+3)^3$$

$$\leq \left(1 - \frac{\beta\mu\ell}{\mu+\ell}\right)\|\tilde{u}_{t,k-1}\|_2^2 - \left(\frac{2}{\beta(\mu+\ell)} - 6\right)\|\nabla_y f(\tilde{x}_{t,k}, \tilde{y}_{t,k}) - \nabla_y f(\tilde{x}_{t,k-1}, \tilde{y}_{t,k-1})\|_2^2$$
$$+ \frac{2}{S_{2,y}}\mathbb{E}[\|H_{\mu_2}(\tilde{x}_{t,k}, \tilde{y}_{t,k}, \omega_i, \xi_i) - H_{\mu_2}(\tilde{x}_{t,k-1}, \tilde{y}_{t,k-1}, \omega_i, \xi_i)\|_2^2 | \mathcal{F}_{t,k}]$$
$$+ 3\mu_2^2 \ell^2 (d_2+3)^3 + \frac{\mu_2^2 \ell(\mu+\ell)}{\beta\mu}(d_2+3)^3$$

$$\overset{(iv)}{\leq} \left(1 - \frac{\beta\mu\ell}{\mu+\ell}\right)\|\tilde{u}_{t,k-1}\|_2^2 + \frac{2}{S_{2,y}}\mathbb{E}[\|H_{\mu_2}(\tilde{x}_{t,k}, \tilde{y}_{t,k}, \omega_i, \xi_i) - H_{\mu_2}(\tilde{x}_{t,k-1}, \tilde{y}_{t,k-1}, \omega_i, \xi_i)\|_2^2 | \mathcal{F}_{t,k}]$$

$$+ 3\mu_2^2\ell^2(d_2+3)^3 + \frac{\mu_2^2\ell(\mu+\ell)}{\beta\mu}(d_2+3)^3$$

$$\overset{(v)}{\leq} \left(1 - \frac{\beta\mu\ell}{\mu+\ell}\right)\|\tilde{u}_{t,k-1}\|_2^2 + \frac{2}{S_{2,y}}\left[2(d_2+4)\ell^2\beta^2\|\tilde{u}_{t,k-1}\|_2^2 + 2\mu_2^2(d_2+6)^3\ell^2\right]$$

$$+ 3\mu_2^2\ell^2(d_2+3)^3 + \frac{\mu_2^2\ell(\mu+\ell)}{\beta\mu}(d_2+3)^3$$

$$= \left(1 - \frac{\beta\mu\ell}{\mu+\ell} + \frac{4}{S_{2,y}}(d_2+4)\ell^2\beta^2\right)\|\tilde{u}_{t,k-1}\|_2^2$$

$$+ \frac{4}{S_{2,y}}\mu_2^2(d_2+6)^3\ell^2 + 3\mu_2^2\ell^2(d_2+3)^3 + \frac{\mu_2^2\ell(\mu+\ell)}{\beta\mu}(d_2+3)^3$$

$$\overset{(vi)}{\leq} \left(1 - \frac{\beta\mu\ell}{2(\mu+\ell)}\right)\|\tilde{u}_{t,k-1}\|_2^2 + \frac{\mu_2^2\ell(1+\kappa)}{\beta}(d_2+3)^3 + 7\mu_2^2(d_2+6)^3\ell^2. \tag{27}$$

where $(i)$ follows from eq. (18) in Lemma 3 and Young's inequality, $(ii)$ follows from Lemma 9, $(iii)$ follows from Lemma 1 in Fang et al. (2018), $(iv)$ follows from the fact that $\frac{2}{\beta(\mu+\ell)} - 6 > 0$, $(v)$ follows from Lemma 11, and $(vi)$ follows from the fact that $\frac{4}{S_{2,y}}(d_2+4)\ell^2\beta^2 \leq \frac{\beta\mu\ell}{2(\mu+\ell)}$. Taking expectation on both sides of eq. (27) and applying eq. (27) iteratively yield

$$\mathbb{E}[\|\tilde{u}_{t,k}\|_2^2] \leq b^k\mathbb{E}[\|\tilde{u}_{t,0}\|_2^2] + \left[\frac{2\mu_2^2\ell\kappa}{\beta}(d_2+3)^3 + 7\mu_2^2(d_2+6)^3\ell^2\right]\sum_{j=0}^{k-1}b^j. \tag{28}$$

Taking summation of eq. (28) over $k = \{0, \cdots m\}$ yields

$$\sum_{k=0}^{m}\mathbb{E}[\|\tilde{u}_{t,k}\|_2^2] \leq \mathbb{E}[\|\tilde{u}_{t,0}\|_2^2]\sum_{k=0}^{m}b^k + \left[\frac{2\mu_2^2\ell\kappa}{\beta}(d_2+3)^3 + 7\mu_2^2(d_2+6)^3\ell^2\right]\sum_{k=0}^{m}\sum_{j=0}^{k-1}b^j$$

$$\leq \frac{1}{1-b}\mathbb{E}[\|\tilde{u}_{t,0}\|_2^2] + \frac{m+1}{1-b}\left[\frac{2\mu_2^2\ell\kappa}{\beta}(d_2+3)^3 + 7\mu_2^2(d_2+6)^3\ell^2\right],$$

which completes the proof. $\qquad\qquad\square$

## C  Proof of Theorem 1

Following steps similar to those in Lemmas 12-14, at the $t$-th outer-loop iteration, we obtain the following convergence result for the inner loop:

$$\mathbb{E}[\|\nabla p_\tau(\tilde{w}_t)\|_2^2]$$

$$\leq \frac{2}{\gamma\tau(I+1)}\mathbb{E}[\|\nabla p_\tau(w_0)\|_2^2] + \mathbb{E}[\|\nabla p_\tau(w_0) - v_0\|_2^2] + \frac{\ell\gamma}{2-\ell\gamma}\mathbb{E}[\|v_0\|_2^2]$$

$$+ \frac{2}{\gamma(I+1)}\left(\frac{\tau^2}{4\mu}\ell^2(d+3)^3 + \tau^2\ell d\right)$$

$$\leq \left(\frac{2}{\gamma\mu(I+1)} + \frac{2\ell\gamma}{2-\ell\gamma}\right)\mathbb{E}[\|\nabla p_\tau(w_0)\|_2^2] + \left(1 + \frac{2\ell\gamma}{2-\ell\gamma}\right)\mathbb{E}[\|\nabla p_\tau(w_0) - v_0\|_2^2]$$

$$+ \frac{2}{\gamma(I+1)}\left(\frac{\tau^2}{4\mu}\ell^2(d+3)^3 + \tau^2\ell d\right). \tag{29}$$

Then, following steps similar to those in Lemma 10, we can obtain

$$\mathbb{E}[\|\nabla p_\tau(w_0) - v_0\|_2^2] \leq \frac{2\sigma^2}{B_1} + \frac{d\ell^2\delta^2}{2} + \frac{\tau^2}{2}\ell^2(d+3)^3. \tag{30}$$

Letting $\gamma = \frac{2}{9\ell}$, $I = 36\kappa - 1$, substituting eq. (30) into eq. (29), and recalling the fact that $w_I = \tilde{w}_t$ and $w_0 = \tilde{w}_{t-1}$ yield

$$\mathbb{E}[\|\nabla p_\tau(\tilde{w}_t)\|_2^2] \leq \frac{1}{2}\mathbb{E}[\|\nabla p_\tau(\tilde{w}_{t-1})\|_2^2] + \frac{5\sigma^2}{2B_1} + \frac{5d\ell^2\delta^2}{8} + \frac{11\tau^2}{16}\ell^2(d+3)^3 + \frac{\tau^2}{4}\ell\mu d. \tag{31}$$

Applying eq. (31) iteratively from $t = T$ to $0$ yields

$$\mathbb{E}[\|\nabla p_\tau(\tilde{w}_T)\|_2^2] \leq \frac{1}{2^T}\|\nabla p_\tau(\tilde{w}_0)\|_2^2 + \frac{5\sigma^2}{2B_1}\sum_{t=0}^{T-1}\frac{1}{2^t}$$

$$+ \left(\frac{5d\ell^2\delta^2}{8} + \frac{11\tau^2}{16}\ell^2(d+3)^3 + \frac{\tau^2}{4}\ell\mu d\right)\sum_{t=0}^{T-1}\frac{1}{2^t}$$

$$\leq \frac{1}{2^T}\|\nabla p_\tau(\tilde{w}_0)\|_2^2 + \frac{5\sigma^2}{B_1} + \frac{5d\ell^2\delta^2}{4} + \frac{11\tau^2}{8}\ell^2(d+3)^3 + \frac{\tau^2}{2}\ell\mu d. \tag{32}$$

Letting $T = \log_2\frac{5\|\nabla p_\tau(\tilde{w}_0)\|_2^2}{\epsilon}$, $B_1 = \frac{25\sigma^2}{\epsilon}$, $\delta = \frac{2\epsilon^{0.5}}{5\ell d^{0.5}}$, and $\tau = \min\{\frac{\epsilon^{0.5}}{3\ell(d+3)^{1.5}}, \sqrt{\frac{2\epsilon}{5\ell\mu d}}\}$, we have

$$\mathbb{E}[\|\nabla p_\tau(\tilde{w}_T)\|_2^2] \leq \epsilon.$$

The total sample complexity is given by

$$T \cdot (I \cdot B_2 + d \cdot B_1) = \mathcal{O}\left(d\left(\kappa + \frac{1}{\epsilon}\right)\log\left(\frac{1}{\epsilon}\right)\right).$$

**Extension to the finite-sum case:** ZO-iSARAH in Algorithm 2 is also applicable to strongly-convex optimization in the finite-sum case, where the objective function takes the form given by

$$\min_{w\in\mathbb{R}^d} p(w) \triangleq \frac{1}{n}\sum_{i=1}^{n} P(w; \xi_i). \tag{33}$$

To solve the problem in eq. (33), we slightly modify Algorithm 2 by replacing line 5 with the full gradient. Following steps similar to those from eq. (29) to eq. (32), we have

$$\mathbb{E}[\|\nabla p_\tau(\tilde{w}_T)\|_2^2] \leq \frac{1}{2^T}\|\nabla p_\tau(\tilde{w}_0)\|_2^2 + \frac{5d\ell^2\delta^2}{4} + \frac{11\tau^2}{8}\ell^2(d+3)^3 + \frac{\tau^2}{2}\ell\mu d.$$

Letting $T = \log_2\frac{4\|\nabla p_\tau(\tilde{w}_0)\|_2^2}{\epsilon}$, $\delta = \frac{\epsilon^{0.5}}{3\ell d^{0.5}}$, and $\tau = \min\{\frac{\epsilon^{0.5}}{3\ell(d+3)^{1.5}}, \sqrt{\frac{\epsilon}{2\ell\mu d}}\}$, we have

$$\mathbb{E}[\|\nabla p_\tau(\tilde{w}_T)\|_2^2] \leq \epsilon.$$

The total sample complexity is given by

$$T \cdot (I \cdot B_2 + d \cdot n) = \mathcal{O}\left(d\left(\kappa + n\right)\log\left(\frac{1}{\epsilon}\right)\right). \tag{34}$$

Let $P(\cdot; \xi) = -F(x_0, \cdot; \xi)$. Then we can conclude that the sample complexity for the initialization of Algorithm 1 is given by $\mathcal{O}\left(d_2\kappa\log\left(\kappa\right)\right)$ in the online case, and is given by $\mathcal{O}\left(d_2(\kappa + n)\log\left(\kappa\right)\right)$ in the finite-sum case.

## D Proof of Theorem 2

### D.1 Proof of Supporting Lemmas

We define $\Delta_t' = \mathbb{E}[\|\nabla_x f_{\mu_1}(x_t, y_t) - v_t\|_2^2] + \mathbb{E}[\|\nabla_y f_{\mu_2}(x_t, y_t) - u_t\|_2^2]$, $\widetilde{\Delta}_{t,k}' = \mathbb{E}[\|\nabla_x f_{\mu_1}(\tilde{x}_{t,k}, \tilde{y}_{t,k}) - \tilde{v}_{t,k}\|_2^2] + \mathbb{E}[\|\nabla_y f_{\mu_2}(\tilde{x}_{t,k}, \tilde{y}_{t,k}) - \tilde{u}_{t,k}\|_2^2]$, and $\delta_t' = \mathbb{E}[\|\nabla_y f_{\mu_2}(x_t, y_t)\|_2^2]$. In this subsection, we establish the following lemmas to characterize the relationship between $\Delta_t$ and $\Delta_t'$, and $\delta_t$ and $\delta_t'$, and the recursive relationship of $\Delta_t'$ and $\delta_t'$, which are crucial for the analysis of Theorem 2.

**Lemma 16.** *Suppose Assumption 2 holds. Then, for any $0 \le t \le T - 1$, we have*

$$\Delta_t \le 2\Delta_t' + \frac{\mu_1^2}{2}\ell^2(d_1 + 3)^3 + \frac{\mu_2^2}{2}\ell^2(d_2 + 3)^3,$$

*and*

$$\delta_t \le 2\delta_t' + \frac{\mu_2^2}{2}\ell^2(d_2 + 3)^3.$$

*Proof.* For the first inequality, we have

$$
\begin{aligned}
\Delta_t &= \mathbb{E}[\|\nabla_x f(x_t, y_t) - v_t\|_2^2] + \mathbb{E}[\|\nabla_y f(x_t, y_t) - u_t\|_2^2] \\
&= \mathbb{E}[\|\nabla_x f_{\mu_1}(x_t, y_t) - v_t + \nabla_x f(x_t, y_t) - \nabla_x f_{\mu_1}(x_t, y_t)\|_2^2] \\
&\quad + \mathbb{E}[\|\nabla_y f_{\mu_2}(x_t, y_t) - u_t + \nabla_y f(x_t, y_t) - \nabla_y f_{\mu_2}(x_t, y_t)\|_2^2] \\
&\le 2\mathbb{E}[\|\nabla_x f_{\mu_1}(x_t, y_t) - v_t\|_2^2] + 2\mathbb{E}[\|\nabla_y f_{\mu_2}(x_t, y_t) - u_t\|_2^2] \\
&\quad + 2\mathbb{E}[\|\nabla_x f(x_t, y_t) - \nabla_x f_{\mu_1}(x_t, y_t)\|_2^2] + 2\mathbb{E}[\|\nabla_y f(x_t, y_t) - \nabla_y f_{\mu_2}(x_t, y_t)\|_2^2] \\
&\overset{(i)}{\le} 2\Delta_t' + \frac{\mu_1^2}{2}\ell^2(d_1 + 3)^3 + \frac{\mu_2^2}{2}\ell^2(d_2 + 3)^3,
\end{aligned}
$$

where $(i)$ follows from Lemma 9. For the second inequality, we have

$$
\begin{aligned}
\delta_t &= \mathbb{E}[\|\nabla_y f(x_t, y_t)\|_2^2] = \mathbb{E}[\|\nabla_y f_{\mu_2}(x_t, y_t) + \nabla_y f(x_t, y_t) - \nabla_y f_{\mu_2}(x_t, y_t)\|_2^2] \\
&\le 2\mathbb{E}[\|\nabla_y f_{\mu_2}(x_t, y_t)\|_2^2] + 2\mathbb{E}[\|\nabla_y f(x_t, y_t) - \nabla_y f_{\mu_2}(x_t, y_t)\|_2^2] \\
&\overset{(i)}{\le} 2\delta_t' + \frac{\mu_2^2}{2}\ell^2(d_2 + 3)^3,
\end{aligned}
$$

where $(i)$ follows from Lemma 9. $\qquad \square$

We provide the following two lemmas to characterize the relationship between $\delta_t'$ and $\delta_{t-1}'$ as well as that between $\Delta_t'$ and $\Delta_{t-1}'$.

**Lemma 17.** *Suppose Assumption 2 holds. Then, we have*

$$
\begin{aligned}
\Delta_t' &\le \left[1 + \frac{6\ell^2\beta^2}{1 - b}\left(\frac{d_1 + 4}{S_{2,x}} + \frac{d_2 + 4}{S_{2,y}}\right)\right]\Delta_{t-1}' + \frac{6\ell^2\beta^2}{1 - b}\left(\frac{d_1 + 4}{S_{2,x}} + \frac{d_2 + 4}{S_{2,y}}\right)\delta_{t-1}' \\
&\quad + 2\ell^2\alpha^2\left(\frac{d_1 + 4}{S_{2,x}} + \frac{d_2 + 4}{S_{2,y}}\right)\left(1 + \frac{9\ell^2\beta^2}{1 - b}\right)\mathbb{E}[\|v_{t-1}\|_2^2] + \pi_\Delta(d_1, d_2, \mu_1, \mu_2),
\end{aligned}
$$

*where $b = 1 - \frac{\beta\mu\ell}{2(\mu+\ell)}$ and*

$$
\begin{aligned}
&\pi_\Delta(d_1, d_2, \mu_1, \mu_2) \\
&= \frac{2\ell^2\beta^2}{1 - b}\left(\frac{d_1 + 4}{S_{2,x}} + \frac{d_2 + 4}{S_{2,y}}\right)\left\{6\ell^2\left[\frac{\mu_1^2(d_1 + 6)^3}{S_{2,x}} + \frac{\mu_2^2(d_2 + 6)^3}{S_{2,y}}\right] + (m+1)\left(\frac{2\mu_2^2\ell\kappa}{\beta}(d_2 + 3)^3\right. \right. \\
&\quad \left. \left. + 7\mu_2^2(d_2 + 6)^3\ell^2\right)\right\} + \frac{2(m+2)\mu_1^2(d_1 + 6)^3\ell^2}{S_{2,x}} + \frac{2(m+2)\mu_2^2(d_2 + 6)^3\ell^2}{S_{2,y}}.
\end{aligned}
$$

*Moreover, if we let $\beta = \frac{2}{13\ell}$, $m = 104\kappa - 1$, $S_{2,x} \ge 5600(d_1 + 4)$ and $S_{2,y} \ge 5600(d_2 + 4)$, then we have*

$$\pi_\Delta(d_1, d_2, \mu_1, \mu_2) \le \kappa^3\ell^2[\mu_1^2(d_1 + 6)^3 + \mu_2^2(d_2 + 6)^3].$$

*Proof.* We proceed as follows:

$$\Delta'_t$$
$$= \widetilde{\Delta}'_{t-1,\bar{m}_{t-1}}$$
$$= \mathbb{E}\Big[ \big\| \nabla_x f_{\mu_1}(\tilde{x}_{t-1,\tilde{m}_{t-1}}, \tilde{y}_{t-1,\tilde{m}_{t-1}}) - \tilde{v}_{t-1,\tilde{m}_{t-1}} \big\|_2^2 \Big]$$
$$\overset{(i)}{\leq} \mathbb{E}\Big[ \big\| \nabla_x f_{\mu_1}(\tilde{x}_{t-1,\tilde{m}_{t-1}-1}, \tilde{y}_{t-1,\tilde{m}_{t-1}-1}) - \tilde{v}_{t-1,\tilde{m}_{t-1}-1} \big\|_2^2 \Big]$$
$$+ \frac{1}{S_{2,x}} \mathbb{E}\Big[ \big\| G_{\mu_1}(\tilde{x}_{t-1,\tilde{m}_{t-1}}, \tilde{y}_{t-1,\tilde{m}_{t-1}}, \nu_i, \xi_i) - G_{\mu_1}(\tilde{x}_{t-1,\tilde{m}_{t-1}-1}, \tilde{y}_{t-1,\tilde{m}_{t-1}-1}, \nu_i, \xi_i) \big\|_2^2 \Big]$$
$$\overset{(ii)}{\leq} \mathbb{E}\Big[ \big\| \nabla_x f_{\mu_1}(\tilde{x}_{t-1,\tilde{m}_{t-1}-1}, \tilde{y}_{t-1,\tilde{m}_{t-1}-1}) - \tilde{v}_{t-1,\tilde{m}_{t-1}-1} \big\|_2^2 \Big]$$
$$+ \frac{1}{S_{2,x}} \Big[ 2(d_1+4)\ell^2\beta^2 \mathbb{E}[\big\| \tilde{u}_{t-1,\tilde{m}_{t-1}-1} \big\|_2^2] + 2\mu_1^2(d_1+6)^3\ell^2 \Big], \tag{35}$$

where $(i)$ follows from Lemma 5, and $(ii)$ follows from Lemma 11. Applying eq. (35) recursively yields

$$\mathbb{E}\Big[ \big\| \nabla_x f_{\mu_1}(\tilde{x}_{t-1,\tilde{m}_{t-1}}, \tilde{y}_{t-1,\tilde{m}_{t-1}}) - \tilde{v}_{t-1,\tilde{m}_{t-1}} \big\|_2^2 \Big]$$
$$\leq \mathbb{E}\Big[ \big\| \nabla_x f_{\mu_1}(\tilde{x}_{t-1,0}, \tilde{y}_{t-1,0}) - \tilde{v}_{t-1,0} \big\|_2^2 \Big] + \frac{2(d_1+4)\ell^2\beta^2}{S_{2,x}} \sum_{k=0}^{\tilde{m}_{t-1}-1} \mathbb{E}[\big\| \tilde{u}_{t-1,k} \big\|_2^2]$$
$$+ \frac{2\tilde{m}_{t-1}\mu_1^2(d_1+6)^3\ell^2}{S_{2,x}}$$
$$\leq \mathbb{E}\Big[ \big\| \nabla_x f_{\mu_1}(\tilde{x}_{t-1,0}, \tilde{y}_{t-1,0}) - \tilde{v}_{t-1,0} \big\|_2^2 \Big] + \frac{2(d_1+4)\ell^2\beta^2}{S_{2,x}} \sum_{k=0}^{m} \mathbb{E}[\big\| \tilde{u}_{t-1,k} \big\|_2^2]$$
$$+ \frac{2(m+1)\mu_1^2(d_1+6)^3\ell^2}{S_{2,x}}. \tag{36}$$

Similarly, we obtain

$$\mathbb{E}\Big[ \big\| \nabla_y f_{\mu_2}(\tilde{x}_{t-1,\tilde{m}_{t-1}}, \tilde{y}_{t-1,\tilde{m}_{t-1}}) - \tilde{u}_{t-1,\tilde{m}_{t-1}} \big\|_2^2 \Big]$$
$$\leq \mathbb{E}\Big[ \big\| \nabla_y f_{\mu_2}(\tilde{x}_{t-1,0}, \tilde{y}_{t-1,0}) - \tilde{u}_{t-1,0} \big\|_2^2 \Big] + \frac{2(d_2+4)\ell^2\beta^2}{S_{2,y}} \sum_{k=0}^{m} \mathbb{E}[\big\| \tilde{u}_{t-1,k} \big\|_2^2]$$
$$+ \frac{2(m+1)\mu_2^2(d_2+6)^3\ell^2}{S_{2,y}}. \tag{37}$$

Combining eq. (36) and eq. (37) yields

$$\Delta'_t \leq \widetilde{\Delta}'_{t-1,0} + \Big( \frac{2(d_1+4)\ell^2\beta^2}{S_{2,x}} + \frac{2(d_2+4)\ell^2\beta^2}{S_{2,y}} \Big) \sum_{k=0}^{m} \mathbb{E}[\big\| \tilde{u}_{t-1,k} \big\|_2^2]$$
$$+ \frac{2(m+1)\mu_1^2(d_1+6)^3\ell^2}{S_{2,x}} + \frac{2(m+1)\mu_2^2(d_2+6)^3\ell^2}{S_{2,y}}. \tag{38}$$

For $\widetilde{\Delta}'_{t-1,0}$, we obtain

$$\widetilde{\Delta}'_{t-1,0} = \mathbb{E}[\|\nabla_x f_{\mu_1}(\tilde{x}_{t-1,0}, \tilde{y}_{t-1,0}) - \tilde{v}_{t-1,0}\|_2^2] + \mathbb{E}[\|\nabla_y f_{\mu_2}(\tilde{x}_{t-1,0}, \tilde{y}_{t-1,0}) - \tilde{u}_{t-1,0}\|_2^2]$$
$$\overset{(i)}{\leq} \mathbb{E}[\|\nabla_x f_{\mu_1}(\tilde{x}_{t-1,-1}, \tilde{y}_{t-1,-1}) - \tilde{v}_{t-1,-1}\|_2^2] + \mathbb{E}[\|\nabla_y f_{\mu_2}(\tilde{x}_{t-1,-1}, \tilde{y}_{t-1,-1}) - \tilde{u}_{t-1,-1}\|_2^2]$$
$$+ \frac{1}{S_{2,x}} \mathbb{E}[\|G(\tilde{x}_{t,0}, \tilde{y}_{t,0}, \nu_i, \xi_i) - G(\tilde{x}_{t,-1}, \tilde{y}_{t,-1}, \nu_{\mathcal{M}_i}, \xi_i)\|_2^2]$$

$$+ \frac{1}{S_{2,y}} \mathbb{E}[\| H(\tilde{x}_{t,0}, \tilde{y}_{t,0}, \nu_i, \xi_i) - H(\tilde{x}_{t,-1}, \tilde{y}_{t,-1}, \nu_{\mathcal{M}_i}, \xi_i) \|_2^2]$$

$$\overset{(ii)}{\leq} \Delta'_{t-1} + \left( \frac{2(d_1+4)\ell^2\alpha^2}{S_{2,x}} + \frac{2(d_2+4)\ell^2\alpha^2}{S_{2,y}} \right) \mathbb{E}[\|v_{t-1}\|_2^2]$$

$$+ \frac{2\mu_1^2(d_1+6)^3\ell^2}{S_{2,x}} + \frac{2\mu_2^2(d_2+6)^3\ell^2}{S_{2,y}}, \tag{39}$$

where $(i)$ follows from Lemma 5 and $(ii)$ follows from Lemma 11. Substituting eq. (39) into eq. (38) yields

$$\Delta'_t \leq \Delta'_{t-1} + \left( \frac{2(d_1+4)\ell^2\alpha^2}{S_{2,x}} + \frac{2(d_2+4)\ell^2\alpha^2}{S_{2,y}} \right) \mathbb{E}[\|v_{t-1}\|_2^2]$$

$$+ \left( \frac{2(d_1+4)\ell^2\beta^2}{S_{2,x}} + \frac{2(d_2+4)\ell^2\beta^2}{S_{2,y}} \right) \sum_{k=0}^{m} \mathbb{E}[\|\tilde{u}_{t-1,k}\|_2^2]$$

$$+ \frac{2(m+2)\mu_1^2(d_1+6)^3\ell^2}{S_{2,x}} + \frac{2(m+2)\mu_2^2(d_2+6)^3\ell^2}{S_{2,y}}$$

$$\overset{(i)}{\leq} \Delta'_{t-1} + \left( \frac{2(d_1+4)\ell^2\alpha^2}{S_{2,x}} + \frac{2(d_2+4)\ell^2\alpha^2}{S_{2,y}} \right) \mathbb{E}[\|v_{t-1}\|_2^2]$$

$$+ \frac{2(m+2)\mu_1^2(d_1+6)^3\ell^2}{S_{2,x}} + \frac{2(m+2)\mu_2^2(d_2+6)^3\ell^2}{S_{2,y}}$$

$$+ \frac{2\ell^2\beta^2}{1-b} \left( \frac{d_1+4}{S_{2,x}} + \frac{d_2+4}{S_{2,y}} \right) \left[ \mathbb{E}[\|\tilde{u}_{t,0}\|_2^2] + (m+1)\left( \frac{2\mu_2^2\ell\kappa}{\beta}(d_2+3)^3 + 7\mu_2^2(d_2+6)^3\ell^2 \right) \right]. \tag{40}$$

where $(i)$ follows from Lemma 15. We next bound the term $\mathbb{E}[\|\tilde{u}_{t-1,0}\|_2^2]$ as follows:

$$\mathbb{E}[\|\tilde{u}_{t-1,0}\|_2^2]$$

$$= \mathbb{E}[\|\tilde{u}_{t-1,0} - \nabla_y f_{\mu_2}(x_t, y_{t-1}) + \nabla_y f_{\mu_2}(x_t, y_{t-1}) - \nabla_y f_{\mu_2}(x_{t-1}, y_{t-1}) + \nabla_y f_{\mu_2}(x_{t-1}, y_{t-1})\|_2^2]$$

$$\leq 3\mathbb{E}[\|\tilde{u}_{t-1,0} - \nabla_y f_{\mu_2}(x_t, y_{t-1})\|_2^2] + 3\mathbb{E}[\|\nabla_y f_{\mu_2}(x_t, y_{t-1}) - \nabla_y f_{\mu_2}(x_{t-1}, y_{t-1})\|_2^2]$$

$$+ 3\mathbb{E}[\|\nabla_y f_{\mu_2}(x_{t-1}, y_{t-1})\|_2^2]$$

$$\overset{(i)}{\leq} 3\mathbb{E}[\|\tilde{u}_{t-1,0} - \nabla_y f_{\mu_2}(x_t, y_{t-1})\|_2^2] + 3\ell^2\mathbb{E}[\|x_t - x_{t-1}\|_2^2] + 3\delta'_{t-1}$$

$$= 3\mathbb{E}[\|\tilde{u}_{t-1,0} - \nabla_y f_{\mu_2}(\tilde{x}_{t-1,0}, \tilde{y}_{t-1,0})\|_2^2] + 3\alpha^2\ell^2\mathbb{E}[\|v_{t-1}\|_2^2] + 3\delta'_{t-1}$$

$$\leq 3\widetilde{\Delta}'_{t-1,0} + 3\alpha^2\ell^2\mathbb{E}[\|v_{t-1}\|_2^2] + 3\delta'_{t-1}$$

$$\overset{(ii)}{\leq} 3\Delta'_{t-1} + 3\delta'_{t-1} + \left[ 3 + \frac{6(d_1+4)}{S_{2,x}} + \frac{6(d_2+4)}{S_{2,y}} \right] \alpha^2\ell^2\mathbb{E}[\|v_{t-1}\|_2^2] + 6\ell^2 \left[ \frac{\mu_1^2(d_1+6)^3}{S_{2,x}} + \frac{\mu_2^2(d_2+6)^3}{S_{2,y}} \right]$$

$$\overset{(iii)}{\leq} 3\Delta'_{t-1} + 3\delta'_{t-1} + 9\alpha^2\ell^2\mathbb{E}[\|v_{t-1}\|_2^2] + 6\ell^2 \left[ \frac{\mu_1^2(d_1+6)^3}{S_{2,x}} + \frac{\mu_2^2(d_2+6)^3}{S_{2,y}} \right] \tag{41}$$

where $(i)$ follows from Lemma 7, and $(ii)$ follows from eq. (39), and $(iii)$ follows from the fact that $S_{2,x} \geq 2(d_1+4)$ and $S_{2,y} \geq 2(d_2+4)$. Substituting eq. (41) into eq. (40) yields

$$\Delta'_t \leq \left[ 1 + \frac{6\ell^2\beta^2}{1-b} \left( \frac{d_1+4}{S_{2,x}} + \frac{d_2+4}{S_{2,y}} \right) \right] \Delta'_{t-1} + \frac{6\ell^2\beta^2}{1-b} \left( \frac{d_1+4}{S_{2,x}} + \frac{d_2+4}{S_{2,y}} \right) \delta'_{t-1}$$

$$+ 2\ell^2\alpha^2 \left( \frac{d_1+4}{S_{2,x}} + \frac{d_2+4}{S_{2,y}} \right) \left( 1 + \frac{9\ell^2\beta^2}{1-b} \right) \mathbb{E}[\|v_{t-1}\|_2^2]$$

$$+ \frac{2\ell^2\beta^2}{1-b} \left( \frac{d_1+4}{S_{2,x}} + \frac{d_2+4}{S_{2,y}} \right) \left\{ 6\ell^2 \left[ \frac{\mu_1^2(d_1+6)^3}{S_{2,x}} + \frac{\mu_2^2(d_2+6)^3}{S_{2,y}} \right] + (m+1)\left( \frac{2\mu_2^2\ell\kappa}{\beta}(d_2+3)^3 \right. \right.$$

$$\left. \left. + 7\mu_2^2(d_2+6)^3\ell^2 \right) \right\} + \frac{2(m+2)\mu_1^2(d_1+6)^3\ell^2}{S_{2,x}} + \frac{2(m+2)\mu_2^2(d_2+6)^3\ell^2}{S_{2,y}}$$

$$\overset{(i)}{\leq} \left[1 + \frac{6\ell^2\beta^2}{1-b}\left(\frac{d_1+4}{S_{2,x}} + \frac{d_2+4}{S_{2,y}}\right)\right]\Delta'_{t-1} + \frac{6\ell^2\beta^2}{1-b}\left(\frac{d_1+4}{S_{2,x}} + \frac{d_2+4}{S_{2,y}}\right)\delta'_{t-1}$$

$$+ 2\ell^2\alpha^2\left(\frac{d_1+4}{S_{2,x}} + \frac{d_2+4}{S_{2,y}}\right)\left(1 + \frac{9\ell^2\beta^2}{1-b}\right)\mathbb{E}[\|v_{t-1}\|_2^2] + \pi_\Delta(d_1, d_2, \mu_1, \mu_2), \tag{42}$$

where $(i)$ follows from the definition of $\pi_\Delta$. $\qquad\square$

**Lemma 18.** *Suppose Assumptions 2-3 hold. Let $S_{2,x} \geq 2d_1 + 8$ and $S_{2,y} \geq 2d_1 + 8$. Then, we have*

$$\delta'_t \leq \left(\frac{4}{\beta\mu(m+1)} + \frac{3\ell\beta}{2-\ell\beta}\right)\delta'_{t-1} + \frac{2+2\ell\beta}{2-\ell\beta}\Delta'_{t-1} + \left(\frac{4\ell^2\alpha^2}{\beta\mu(m+1)} + 2\ell^2\alpha^2 + \frac{9\ell^3\beta\alpha^2}{2-\ell\beta}\right)\mathbb{E}[\|v_{t-1}\|_2^2]$$

$$+ \pi_\delta(d_1, d_2, \mu_1, \mu_2),$$

*where*

$$\pi_\delta(d_1, d_2, \mu_1, \mu_2) = \frac{2\ell^2(2+2\ell\beta)}{2-\ell\beta}\left(\frac{\mu_1^2(d_1+6)^3}{S_{2,x}} + \frac{\mu_2^2(d_2+6)^3}{S_{2,y}}\right) + \frac{2}{\beta(m+1)}\left(\frac{\mu_2^2}{4\mu}\ell^2(d_2+3)^3 + \mu_2^2\ell d_2\right).$$

*Furthermore, if we let $\beta = \frac{2}{13\ell}$, $m = 104\kappa - 1$, then we have*

$$\pi_\delta(d_1, d_2, \mu_1, \mu_2) = \frac{5}{2}\mu_1^2\ell^2(d_1+6)^3 + 3\mu_2^2\ell^2(d_2+6)^3 + \frac{1}{8}\mu_2^2\mu\ell d_2.$$

*Proof.* Using the result in Lemma 14, and recalling the definition in Appendix B.2 that $\nabla g_{t,\mu_2}(\tilde{y}_{t,\tilde{m}_t}) = \nabla_y f(x_t, y_t)$ and $\nabla g_{t,\mu_2}(\tilde{y}_{t,0}) = \nabla_y f_{\mu_2}(x_{t+1}, y_t)$, we have

$$\delta'_{t+1} \leq \frac{2}{\beta\mu(m+1)}\mathbb{E}[\|\nabla_y f_{\mu_2}(x_{t+1}, y_t)\|_2^2] + \mathbb{E}[\|\nabla g_{t,\mu_2}(\tilde{y}_{t,0}) - \tilde{u}_{t,0}\|_2^2] + \frac{\ell\beta}{2-\ell\beta}\mathbb{E}[\|\tilde{u}_{t,0}\|_2^2]$$

$$+ \frac{2}{\beta(m+1)}\left(\frac{\mu_2^2}{4\mu}\ell^2(d_2+3)^3 + \mu_2^2\ell d_2\right)$$

$$\leq \frac{2}{\beta\mu(m+1)}\mathbb{E}[\|\nabla_y f_{\mu_2}(x_{t+1}, y_t)\|_2^2] + \widetilde{\Delta}'_{t,0} + \frac{\ell\beta}{2-\ell\beta}\mathbb{E}[\|\tilde{u}_{t,0}\|_2^2]$$

$$+ \frac{2}{\beta(m+1)}\left(\frac{\mu_2^2}{4\mu}\ell^2(d_2+3)^3 + \mu_2^2\ell d_2\right)$$

$$\leq \frac{4}{\beta\mu(m+1)}\mathbb{E}[\|\nabla_y f_{\mu_2}(x_{t+1}, y_t) - \nabla_y f_{\mu_2}(x_t, y_t)\|_2^2] + \frac{4}{\beta\mu(m+1)}\mathbb{E}[\|\nabla_y f_{\mu_2}(x_t, y_t)\|_2^2]$$

$$+ \widetilde{\Delta}'_{t,0} + \frac{\ell\beta}{2-\ell\beta}\mathbb{E}[\|\tilde{u}_{t,0}\|_2^2] + \frac{2}{\beta(m+1)}\left(\frac{\mu_2^2}{4\mu}\ell^2(d_2+3)^3 + \mu_2^2\ell d_2\right)$$

$$\leq \frac{4\ell^2\alpha^2}{\beta\mu(m+1)}\mathbb{E}[\|v_t\|_2^2] + \frac{4}{\beta\mu(m+1)}\delta'_t + \widetilde{\Delta}'_{t,0} + \frac{\ell\beta}{2-\ell\beta}\mathbb{E}[\|\tilde{u}_{t,0}\|_2^2]$$

$$+ \frac{2}{\beta(m+1)}\left(\frac{\mu_2^2}{4\mu}\ell^2(d_2+3)^3 + \mu_2^2\ell d_2\right)$$

$$\overset{(i)}{\leq} \frac{4\ell^2\alpha^2}{\beta\mu(m+1)}\mathbb{E}[\|v_t\|_2^2] + \frac{4}{\beta\mu(m+1)}\delta'_t$$

$$+ \Delta'_t + 2\ell^2\alpha^2\mathbb{E}[\|v_t\|_2^2] + 2\ell^2\left(\frac{\mu_1^2(d_1+6)^3}{S_{2,x}} + \frac{\mu_2^2(d_2+6)^3}{S_{2,y}}\right)$$

$$+ \frac{\ell\beta}{2-\ell\beta}\left[3\Delta'_t + 3\delta'_t + 9\ell^2\alpha^2\mathbb{E}[\|v_t\|_2^2] + 6\ell^2\left(\frac{\mu_1^2(d_1+6)^3}{S_{2,x}} + \frac{\mu_2^2(d_2+6)^3}{S_{2,y}}\right)\right]$$

$$+ \frac{2}{\beta(m+1)}\left(\frac{\mu_2^2}{4\mu}\ell^2(d_2+3)^3 + \mu_2^2\ell d_2\right)$$

$$= \left(\frac{4}{\beta\mu(m+1)} + \frac{3\ell\beta}{2-\ell\beta}\right)\delta'_t + \frac{2+2\ell\beta}{2-\ell\beta}\Delta'_t + \left(\frac{4\ell^2\alpha^2}{\beta\mu(m+1)} + 2\ell^2\alpha^2 + \frac{9\ell^3\beta\alpha^2}{2-\ell\beta}\right)\mathbb{E}[\|v_t\|_2^2]$$

$$+ \frac{2\ell^2(2+2\ell\beta)}{2-\ell\beta}\left(\frac{\mu_1^2(d_1+6)^3}{S_{2,x}} + \frac{\mu_2^2(d_2+6)^3}{S_{2,y}}\right) + \frac{2}{\beta(m+1)}\left(\frac{\mu_2^2}{4\mu}\ell^2(d_2+3)^3 + \mu_2^2\ell d_2\right),$$

$$\leq \left(\frac{4}{\beta\mu(m+1)} + \frac{3\ell\beta}{2-\ell\beta}\right)\delta'_t + \frac{2+2\ell\beta}{2-\ell\beta}\Delta'_t + \left(\frac{4\ell^2\alpha^2}{\beta\mu(m+1)} + 2\ell^2\alpha^2 + \frac{9\ell^3\beta\alpha^2}{2-\ell\beta}\right)\mathbb{E}[\|v_t\|_2^2]$$

$$+ \pi_\delta(d_1,d_2,\mu_1,\mu_2), \tag{43}$$

where $(i)$ follows from eq. (39) and eq. (41), and from the fact that $S_{2,x} \geq 2d_1 + 8$ and $S_{2,y} \geq 2d_2 + 8$. The proof is completed by shifting the index in eq. (43) from $t$ to $t-1$. $\qquad\square$

## D.2 Proof of Theorem 2

We first restate Theorem 2 as follows to include the specifics of the parameters.

**Theorem 4** (Restate of Theorem 2 with parameter specifics). *Let Assumptions 1,2,4,and 3 hold and apply ZO-VRGDA in Algorithm 1 to solve the problem in eq. (1) with the following parameters:*

$$\zeta = \frac{1}{\kappa}, \quad \alpha = \frac{1}{24(\kappa+1)\ell}, \quad \beta = \frac{2}{13\ell}, \quad q = \frac{2800\kappa}{13\epsilon(\kappa+1)},$$

$$m = 104\kappa - 1, \quad S_{2,x} = \frac{5600(d_1+4)\kappa}{\epsilon}, \quad S_{2,y} = \frac{5600(d_2+4)\kappa}{\epsilon},$$

$$S_1 = \frac{40320\sigma^2\kappa^2}{\epsilon^2}, \quad T = \max\{1728(\kappa+1)\ell\frac{\Phi(x_0)-\Phi^*}{\epsilon^2}, \frac{810\kappa}{\epsilon^2}\},$$

$$\delta = \frac{\epsilon}{71\kappa\ell\sqrt{d_1+d_2}}, \quad \mu_1 = \frac{\epsilon}{71\kappa^{2.5}\ell(d_1+6)^{1.5}}, \quad \mu_2 = \frac{\epsilon}{71\kappa^{2.5}\ell(d_2+6)^{1.5}}.$$

*Algorithm 1 outputs $\hat{x}$ satisfies that*

$$\mathbb{E}[\|\nabla\Phi(\hat{x})\|_2] \leq \epsilon$$

*with at most $\mathcal{O}((d_1+d_2)\kappa^3\epsilon^{-3})$ function queries.*

*Proof.* By Lemma 1, the objective function $\Phi$ is $L$-smooth, which implies that

$$\Phi(x_{t+1}) \leq \Phi(x_t) - \alpha\langle\nabla_x\Phi(x_t),v_t\rangle + \frac{L\alpha^2}{2}\|v_t\|_2^2$$

$$= \Phi(x_t) - \alpha\langle\nabla_x\Phi(x_t) - v_t, v_t\rangle - \alpha\|v_t\|_2^2 + \frac{L\alpha^2}{2}\|v_t\|_2^2$$

$$\overset{(i)}{\leq} \Phi(x_t) + \frac{\alpha}{2}\|\nabla_x\Phi(x_t)-v_t\|_2^2 + \frac{\alpha}{2}\|v_t\|_2^2 - \alpha\|v_t\|_2^2 + \frac{L\alpha^2}{2}\|v_t\|_2^2$$

$$\leq \Phi(x_t) + \frac{\alpha}{2}\|\nabla_x\Phi(x_t)-v_t\|_2^2 - \left(\frac{\alpha}{2} - \frac{L\alpha^2}{2}\right)\|v_t\|_2^2$$

$$\leq \Phi(x_t) + \alpha\|\nabla_x\Phi(x_t) - \nabla_x f(x_t,y_t)\|_2^2 + \alpha\|\nabla_x f(x_t,y_t) - v_t\|_2^2 - \left(\frac{\alpha}{2} - \frac{L\alpha^2}{2}\right)\|v_t\|_2^2$$

$$\overset{(ii)}{\leq} \Phi(x_t) + \alpha\kappa^2\|\nabla_y f(x_t,y_t)\|_2^2 + \alpha\|\nabla_x f(x_t,y_t) - v_t\|_2^2 - \left(\frac{\alpha}{2} - \frac{L\alpha^2}{2}\right)\|v_t\|_2^2, \tag{44}$$

where $(i)$ follows from the fact that $(-1)\langle\nabla_x\Phi(x_t) - v_t, v_t\rangle \leq \frac{1}{2}\|\nabla_x\Phi(x_t) - v_t\|_2^2 + \frac{1}{2}\|v_t\|_2^2$, and $(ii)$ follows from the fact that

$$\|\nabla_x\Phi(x_t) - \nabla_x f(x_t,y_t)\|_2^2 = \|\nabla_x f(x_t,y^*(x_t)) - \nabla_x f(x_t,y_t)\|_2^2 \leq \ell^2\|y^*(x_t) - y_t\|_2^2$$

$$\overset{eq.~(19)}{\leq} \frac{\ell^2}{\mu^2}\|\nabla_y f(x_t,y^*(x_t)) - \nabla_y f(x_t,y_t)\|_2^2 = \kappa^2\|\nabla_y f(x_t,y_t)\|_2^2.$$

Taking expectation on both sides of eq. (44) yields

$$\mathbb{E}[\Phi(x_{t+1})] \leq \mathbb{E}[\Phi(x_t)] + \alpha\kappa^2\mathbb{E}[\|\nabla_y f(x_t,y_t)\|_2^2] + \alpha\mathbb{E}[\|\nabla_x f(x_t,y_t) - v_t\|_2^2] - \left(\frac{\alpha}{2} - \frac{L\alpha^2}{2}\right)\mathbb{E}[\|v_t\|_2^2]$$

$$\leq \mathbb{E}[\Phi(x_t)] + \alpha\kappa^2\delta_t + \alpha\Delta_t - \left(\frac{\alpha}{2} - \frac{L\alpha^2}{2}\right)\mathbb{E}[\|v_t\|_2^2]. \tag{45}$$

Using the property in Lemma 16, we obtain the following

$$\mathbb{E}[\Phi(x_{t+1})] \leq \mathbb{E}[\Phi(x_t)] + \alpha\kappa^2\delta_t + \alpha\Delta_t - \left(\frac{\alpha}{2} - \frac{L\alpha^2}{2}\right)\mathbb{E}[\|v_t\|_2^2]$$

$$\overset{(i)}{\leq} \mathbb{E}[\Phi(x_t)] + 2\alpha\kappa^2\delta_t' + 2\alpha\Delta_t' - \left(\frac{\alpha}{2} - \frac{L\alpha^2}{2}\right)\mathbb{E}[\|v_t\|_2^2]$$

$$+ \frac{\mu_2\alpha(\kappa^2+1)}{2}\ell^2(d_2+3)^3 + \frac{\mu_1\alpha}{2}\ell^2(d_1+3)^3. \tag{46}$$

Rearranging eq. (46) and taking the summation over $t = \{0, 1, \cdots, T-1\}$ yield

$$\left(\frac{\alpha}{2} - \frac{L\alpha^2}{2}\right)\sum_{t=0}^{T-1}\mathbb{E}[\|v_t\|_2^2] \leq \Phi(x_0) - \mathbb{E}[\Phi(x_T)] + 2\alpha\kappa^2\sum_{t=0}^{T-1}\delta_t' + 2\alpha\sum_{t=0}^{T-1}\Delta_t'$$

$$+ \alpha T\pi(d_1, d_2, \mu_1, \mu_2). \tag{47}$$

Note that in eq. (47) we define

$$\pi(d_1, d_2, \mu_1, \mu_2) = \frac{\mu_2^2(\kappa^2+1)}{2}\ell^2(d_2+3)^3 + \frac{\mu_1^2}{2}\ell^2(d_1+3)^3. \tag{48}$$

Then we proceed to prove Theorem 2/Theorem 4 in the following three steps.

**Step 1.** *We establish the induction relationships for the tracking error and gradient estimation error with respect to the Gaussian smoothed function upon one outer-loop update for ZO-VRGDA. Namely, we develop the relationship between $\delta_t'$ and $\delta_{t-1}'$ as well as that between $\Delta_t'$ and $\Delta_{t-1}'$, which are captured in Lemma 17 and Lemma 18.*

**Step 2.** *Based on Step 1, we provide the bounds on the inter-related accumulative errors $\sum_{t=0}^{T-1}\Delta_t'$ and $\sum_{t=0}^{T-1}\delta_t'$ over the entire execution of the algorithm.*

We first consider $\sum_{t=0}^{T-1}\Delta_t'$, for any $(n_T-1)q \leq t' < T-1$. Applying the inequality in Lemma 17 recursively, we obtain the following bound

$$\Delta_t' \leq \left[1 + \frac{6\ell^2\beta^2}{1-b}\left(\frac{d_1+4}{S_{2,x}} + \frac{d_2+4}{S_{2,y}}\right)\right]\Delta_{t-1}' + \frac{6\ell^2\beta^2}{1-b}\left(\frac{d_1+4}{S_{2,x}} + \frac{d_2+4}{S_{2,y}}\right)\delta_{t-1}'$$

$$+ 2\ell^2\alpha^2\left(\frac{d_1+4}{S_{2,x}} + \frac{d_2+4}{S_{2,y}}\right)\left(1 + \frac{9\ell^2\beta^2}{1-b}\right)\mathbb{E}[\|v_{T-2}\|_2^2] + \pi_\Delta(d_1, d_2, \mu_1, \mu_2, S_2)$$

$$\leq \left[1 + \frac{6\ell^2\beta^2}{1-b}\left(\frac{d_1+4}{S_{2,x}} + \frac{d_2+4}{S_{2,y}}\right)\right]^{t-t'}\Delta_{t'}'$$

$$+ \frac{6\ell^2\beta^2}{1-b}\left(\frac{d_1+4}{S_{2,x}} + \frac{d_2+4}{S_{2,y}}\right)\sum_{p=t'}^{t-1}\left[1 + \frac{6\ell^2\beta^2}{1-b}\left(\frac{d_1+4}{S_{2,x}} + \frac{d_2+4}{S_{2,y}}\right)\right]^{p-t'}\delta_p'$$

$$+ 2\ell^2\alpha^2\left(\frac{d_1+4}{S_{2,x}} + \frac{d_2+4}{S_{2,y}}\right)\left(1 + \frac{9\ell^2\beta^2}{1-b}\right)\sum_{p=t'}^{t-1}\left[1 + \frac{6\ell^2\beta^2}{1-b}\left(\frac{d_1+4}{S_{2,x}} + \frac{d_2+4}{S_{2,y}}\right)\right]^{p-t'}\mathbb{E}[\|v_t\|_2^2]$$

$$+ \pi_\Delta(d_1, d_2, \mu_1, \mu_2, S_2)\sum_{p=t'}^{t-1}\left[1 + \frac{6\ell^2\beta^2}{1-b}\left(\frac{d_1+4}{S_{2,x}} + \frac{d_2+4}{S_{2,y}}\right)\right]^{p-t'}$$

$$\overset{(i)}{\leq} 2\Delta_{t'}' + \frac{6\ell^2\beta^2}{1-b}\left(\frac{d_1+4}{S_{2,x}} + \frac{d_2+4}{S_{2,y}}\right)\sum_{p=t'}^{t-1}\delta_t'$$

$$+ 2\ell^2\alpha^2 \left( \frac{d_1+4}{S_{2,x}} + \frac{d_2+4}{S_{2,y}} \right) \left( 1 + \frac{9\ell^2\beta^2}{1-b} \right) \sum_{p=t'}^{t-1} \mathbb{E}[\|v_t\|_2^2]$$

$$+ 2\pi_\Delta(d_1, d_2, \mu_1, \mu_2, S_2), \tag{49}$$

where $(i)$ follows from the fact that

$$\left[ 1 + \frac{6\ell^2\beta^2}{1-b} \left( \frac{d_1+4}{S_{2,x}} + \frac{d_2+4}{S_{2,y}} \right) \right]^{p-t'}$$

$$\leq \left[ 1 + \frac{6\ell^2\beta^2}{1-b} \left( \frac{d_1+4}{S_{2,x}} + \frac{d_2+4}{S_{2,y}} \right) \right]^q$$

$$\overset{(ii)}{\leq} 1 + \frac{\frac{6q\ell^2\beta^2}{1-b} \left( \frac{d_1+4}{S_{2,x}} + \frac{d_2+4}{S_{2,y}} \right)}{1 - \frac{6(q-1)\ell^2\beta^2}{1-b} \left( \frac{d_1+4}{S_{2,x}} + \frac{d_2+4}{S_{2,y}} \right)} \overset{(iii)}{\leq} 2,$$

where $(ii)$ follows from the Bernoulli's inequality Li & Yeh (2013)

$$(1+c)^r \leq 1 + \frac{rc}{1-(r-1)c} \quad \text{for} \quad c \in \left[ -1, \frac{1}{r-1} \right), r > 1, \tag{50}$$

and $(iii)$ follows from the fact that $q = (1-b) \left( \frac{d_1+4}{S_{2,x}} + \frac{d_2+4}{S_{2,y}} \right)^{-1}$, $\beta = \frac{2}{13\ell}$, $\left( \frac{d_1+4}{S_{2,x}} + \frac{d_2+4}{S_{2,y}} \right) < 1$, and $b = 1 - \frac{\beta\mu\ell}{2(\mu+\ell)}$, which further implies that

$$\frac{\frac{6q\ell^2\beta^2}{1-b} \left( \frac{d_1+4}{S_{2,x}} + \frac{d_2+4}{S_{2,y}} \right)}{1 - \frac{6(q-1)\ell^2\beta^2}{1-b} \left( \frac{d_1+4}{S_{2,x}} + \frac{d_2+4}{S_{2,y}} \right)} \leq \frac{\frac{6q\ell^2\beta^2}{1-b} \left( \frac{d_1+4}{S_{2,x}} + \frac{d_2+4}{S_{2,y}} \right)}{1 - \frac{6q\ell^2\beta^2}{1-b} \left( \frac{d_1+4}{S_{2,x}} + \frac{d_2+4}{S_{2,y}} \right)} = \frac{6\ell^2\beta^2}{1 - 6\ell^2\beta^2} < 1.$$

Letting $t' = (n_T - 1)q$ and taking summation of eq. (49) over $t = \{(n_T-1)q, \cdots, T-1\}$ yield

$$\sum_{t=(n_T-1)q}^{T-1} \Delta_t' \leq 2(T - (n_T-1)q)\Delta_{(n_T-1)q}' + \frac{6\ell^2\beta^2}{1-b} \left( \frac{d_1+4}{S_{2,x}} + \frac{d_2+4}{S_{2,y}} \right) \sum_{t=(n_T-1)q}^{T-1} \sum_{p=(n_T-1)q}^{t-1} \delta_p'$$

$$+ 2\ell^2\alpha^2 \left( \frac{d_1+4}{S_{2,x}} + \frac{d_2+4}{S_{2,y}} \right) \left( 1 + \frac{9\ell^2\beta^2}{1-b} \right) \sum_{t=(n_T-1)q}^{T-1} \sum_{p=(n_T-1)q}^{t-1} \mathbb{E}[\|v_p\|_2^2]$$

$$+ 2(T - (n_T-1)q)\pi_\Delta(d_1, d_2, \mu_1, \mu_2, S_2)$$

$$\overset{(i)}{\leq} 2(T - (n_T-1)q)\epsilon(S_1, \delta) + \frac{6q\ell^2\beta^2}{1-b} \left( \frac{d_1+4}{S_{2,x}} + \frac{d_2+4}{S_{2,y}} \right) \sum_{t=(n_T-1)q}^{T-2} \delta_t'$$

$$+ 2q\ell^2\alpha^2 \left( \frac{d_1+4}{S_{2,x}} + \frac{d_2+4}{S_{2,y}} \right) \left( 1 + \frac{9\ell^2\beta^2}{1-b} \right) \sum_{t=(n_T-1)q}^{T-2} \mathbb{E}[\|v_t\|_2^2]$$

$$+ 2(T - (n_T-1)q)\pi_\Delta(d_1, d_2, \mu_1, \mu_2)$$

$$= 2(T - (n_T-1)q)\epsilon(S_1, \delta) + 6\ell^2\beta^2 \sum_{t=(n_T-1)q}^{T-2} \delta_t'$$

$$+ 2\ell^2\alpha^2(1-b) \left( 1 + \frac{9\ell^2\beta^2}{1-b} \right) \sum_{t=(n_T-1)q}^{T-2} \mathbb{E}[\|v_t\|_2^2]$$

$$+ 2(T - (n_T-1)q)\pi_\Delta(d_1, d_2, \mu_1, \mu_2)$$

$$\leq 2(T - (n_T-1)q)\epsilon(S_1, \delta) + 6\ell^2\beta^2 \sum_{t=(n_T-1)q}^{T-2} \delta_t' + 2\ell^2\alpha^2 \left( 1 + 9\ell^2\beta^2 \right) \sum_{t=(n_T-1)q}^{T-2} \mathbb{E}[\|v_t\|_2^2]$$

$$+ 2(T - (n_T - 1)q)\pi_\Delta(d_1, d_2, \mu_1, \mu_2)$$

$$\overset{(ii)}{\leq} 2(T - (n_T - 1)q)\epsilon(S_1, \delta) + \frac{1}{7}\sum_{t=(n_T-1)q}^{T-2}\delta_t' + 3\ell^2\alpha^2\sum_{t=(n_T-1)q}^{T-2}\mathbb{E}[\|v_t\|_2^2]$$

$$+ 2(T - (n_T - 1)q)\pi_\Delta(d_1, d_2, \mu_1, \mu_2), \tag{51}$$

where $(i)$ follows from the fact that $\Delta_{(n_T-n)q}' \leq \epsilon(S_1, \delta)$ for all $n \leq n_T$ (following from Lemma 4),

$$\sum_{t=(n_T-1)q}^{T-1}\sum_{p=(n_T-1)q}^{t-1}\delta_p' \leq q\sum_{t=(n_T-1)q}^{T-2}\delta_t',$$

and

$$\sum_{t=(n_T-1)q}^{T-1}\sum_{p=(n_T-1)q}^{t-1}\mathbb{E}[\|v_t\|_2^2] \leq q\sum_{t=(n_T-1)q}^{T-2}\mathbb{E}[\|v_t\|_2^2],$$

and $(ii)$ follows because $\beta = \frac{2}{13\ell}$. Applying steps similar to those in eq. (51) for iterations over $t = \{(n_T - n_t)q, \cdots, (n_T - n_t + 1)q - 1\}$ yields

$$\sum_{t=(n_T-n_t)q}^{(n_T-n_t+1)q-1}\Delta_t' \leq 2q\epsilon(S_1, \delta) + \frac{1}{7}\sum_{t=(n_T-n_t)q}^{(n_T-n_t+1)q-1}\delta_t' + 3\ell^2\alpha^2\sum_{t=(n_T-n_t)q}^{(n_T-n_t+1)q-1}\mathbb{E}[\|v_t\|_2^2]$$

$$+ 2q\pi_\Delta(d_1, d_2, \mu_1, \mu_2). \tag{52}$$

Taking summation of eq. (52) over $n = \{2, \cdots, n_T\}$ and combing with eq. (51) yield

$$\sum_{t=0}^{T-1}\Delta_t' \leq 2T\epsilon(S_1, \delta) + \frac{1}{7}\sum_{t=0}^{T-1}\delta_t' + 3\ell^2\alpha^2\sum_{t=0}^{T-1}\mathbb{E}[\|v_t\|_2^2] + 2T\pi_\Delta(d_1, d_2, \mu_1, \mu_2). \tag{53}$$

Then we consider the upper bound on $\sum_{t=0}^{T-1}\delta_t'$. Since $m = \frac{16}{\mu\beta} - 1$ and $\beta = \frac{2}{13\ell}$, Lemma 18 implies

$$\delta_t' \leq \frac{1}{2}\delta_{t-1}' + \frac{5}{4}\Delta_{t-1}' + 3\ell^2\alpha^2\mathbb{E}[\|v_{t-1}\|_2^2] + \pi_\delta(d_1, d_2, \mu_1, \mu_2). \tag{54}$$

Applying eq. (54) recursively from $t$ to 0 yields

$$\delta_t' \leq \frac{1}{2^t}\delta_0' + \frac{5}{4}\sum_{p=0}^{t-1}\frac{1}{2^p}\Delta_p' + 3\ell^2\alpha^2\sum_{p=0}^{t-1}\frac{1}{2^p}\mathbb{E}[\|v_p\|_2^2] + \pi_\delta(d_1, d_2, \mu_1, \mu_2)\sum_{p=0}^{t-1}\frac{1}{2^p}. \tag{55}$$

Taking the summation of eq. (55) over $t = \{0, 1, \cdots, T - 1\}$ yields

$$\sum_{t=0}^{T-1}\delta_t' \leq \delta_0'\sum_{t=0}^{T-1}\frac{1}{2^t} + \frac{5}{4}\sum_{t=0}^{T-1}\sum_{p=0}^{t-1}\frac{1}{2^p}\Delta_p' + 3\ell^2\alpha^2\sum_{t=0}^{T-1}\sum_{p=0}^{t-1}\frac{1}{2^p}\mathbb{E}[\|v_p\|_2^2]$$

$$+ \pi_\delta(d_1, d_2, \mu_1, \mu_2)\sum_{t=0}^{T-1}\sum_{p=0}^{t-1}\frac{1}{2^p}$$

$$\leq 2\delta_0' + \frac{5}{2}\sum_{t=0}^{T-2}\Delta_t' + 6\ell^2\alpha^2\sum_{t=0}^{T-2}\mathbb{E}[\|v_t\|_2^2] + 2T\pi_\delta(d_1, d_2, \mu_1, \mu_2). \tag{56}$$

We then decouple the bounds on $\sum_{t=0}^{T-1}\Delta_t'$ and $\sum_{t=0}^{T-1}\delta_t'$ in Step 2 from each other, and establish their separate relationships with the accumulative gradient estimators $\sum_{i=0}^{T-1}\mathbb{E}[\|v_t\|_2^2]$.

Substituting eq. (56) into eq. (53) yields

$$\sum_{t=0}^{T-1} \Delta_t' \leq 2T\epsilon(S_1, \delta) + \frac{2}{7}\delta_0' + 4\alpha^2\ell^2 \sum_{t=0}^{T-2} \mathbb{E}[\|v_t\|_2^2] + \frac{5}{14}\sum_{t=0}^{T-2} \Delta_t'$$
$$+ 2T\pi_\Delta(d_1, d_2, \mu_1, \mu_2) + \frac{2}{7}T\pi_\delta(d_1, d_2, \mu_1, \mu_2),$$

which implies

$$\sum_{t=0}^{T-1} \Delta_t' \leq 4T\epsilon(S_1, \delta) + \frac{1}{2}\delta_0' + 7\alpha^2\ell^2 \sum_{t=0}^{T-2} \mathbb{E}[\|v_t\|_2^2]$$
$$+ \frac{1}{2}T\pi_\Delta(d_1, d_2, \mu_1, \mu_2) + 4T\pi_\delta(d_1, d_2, \mu_1, \mu_2). \tag{57}$$

Substituting eq. (57) into eq. (56) yields

$$\sum_{t=0}^{T-1} \delta_t' \leq 10T\epsilon(S_1, \delta) + 4\delta_0' + 24\alpha^2\ell^2 \sum_{t=0}^{T-2} \mathbb{E}[\|v_t\|_2^2]$$
$$+ 10T\pi_\Delta(d_1, d_2, \mu_1, \mu_2) + 4T\pi_\delta(d_1, d_2, \mu_1, \mu_2). \tag{58}$$

**Step 3.** *We bound $\sum_{i=0}^{T-1} \mathbb{E}[\|v_t\|_2^2]$, and further cancel out the impact of $\sum_{t=0}^{T-1} \Delta_t'$ and $\sum_{t=0}^{T-1} \delta_t'$ by exploiting Step 2. Then, we obtain the convergence rate of $\mathbb{E}[\|\nabla\Phi(\hat{x})\|_2^2]$.*

Substituting eq. (57) and eq. (58) into eq. (47) yields

$$\left(\frac{\alpha}{2} - \frac{L\alpha^2}{2}\right) \sum_{t=0}^{T-1} \mathbb{E}[\|v_t\|_2^2]$$

$$\leq \Phi(x_0) - \mathbb{E}[\Phi(x_T)] + (20\kappa^2 + 8)\alpha T\epsilon(S_1, \delta) + (8\kappa^2 + 1)\alpha\delta_0' + (48\kappa^2 + 14)\alpha^3\ell^2 \sum_{t=0}^{T-1} \mathbb{E}[\|v_t\|_2^2]$$

$$+ (20\kappa^2 + 1)\alpha T\pi_\Delta(d_1, d_2, \mu_1, \mu_2) + (8\kappa^2 + 8)\alpha T\pi_\delta(d_1, d_2, \mu_1, \mu_2) + \alpha T\pi(d_1, d_2, \mu_1, \mu_2)$$

$$\overset{(i)}{\leq} \Phi(x_0) - \mathbb{E}[\Phi(x_T)] + 28\kappa^2\alpha T\epsilon(S_1, \delta) + 9\kappa^2\alpha\delta_0' + 62\alpha^3 L^2 \sum_{t=0}^{T-1} \mathbb{E}[\|v_t\|_2^2]$$

$$+ 21\kappa^2\alpha T\pi_\Delta(d_1, d_2, \mu_1, \mu_2) + 16\kappa^2\alpha T\pi_\delta(d_1, d_2, \mu_1, \mu_2) + \alpha T\pi(d_1, d_2, \mu_1, \mu_2), \tag{59}$$

where $(i)$ follows from the fact that $L = (1 + \kappa)\ell$ and $\kappa > 1$. Rearranging eq. (59), we have

$$\left(\frac{\alpha}{2} - \frac{L\alpha^2}{2} - 62L^2\alpha^3\right) \sum_{t=0}^{T-1} \mathbb{E}[\|v_t\|_2^2]$$
$$\leq \Phi(x_0) - \mathbb{E}[\Phi(x_T)] + 28\kappa^2\alpha T\epsilon(S_1, \delta) + 9\kappa^2\alpha\delta_0'$$
$$+ 21\kappa^2\alpha T\pi_\Delta(d_1, d_2, \mu_1, \mu_2) + 16\kappa^2\alpha T\pi_\delta(d_1, d_2, \mu_1, \mu_2) + \alpha T\pi(d_1, d_2, \mu_1, \mu_2). \tag{60}$$

Since $\alpha = \frac{1}{24L}$, we obtain

$$\frac{\alpha}{2} - \frac{L\alpha^2}{2} - 62L^2\alpha^3 = \frac{214}{13824L} \geq \frac{1}{72L}. \tag{61}$$

Substituting eq. (61) into eq. (60) and applying Assumption 1 yield

$$\sum_{t=0}^{T-1} \mathbb{E}[\|v_t\|_2^2] \leq 72L(\Phi(x_0) - \Phi^*) + 84\kappa^2 T\epsilon(S_1, \delta) + 27\kappa^2\delta_0'$$

$$+ 63\kappa^2 T \pi_\Delta(d_1, d_2, \mu_1, \mu_2) + 48\kappa^2 T \pi_\delta(d_1, d_2, \mu_1, \mu_2)$$
$$+ 3T\pi(d_1, d_2, \mu_1, \mu_2). \tag{62}$$

We then establish the convergence bound on $\mathbb{E}[\|\nabla\Phi(\hat{x})\|_2]$ based on the bounds on its estimators $\sum_{i=0}^{T-1} \mathbb{E}[\|v_t\|_2^2]$ and the two error bounds $\sum_{t=0}^{T-1} \Delta'_t$, and $\sum_{t=0}^{T-1} \delta'_t$.

Observe that

$$\sum_{t=0}^{T-1} \mathbb{E}[\|\nabla\Phi(x_t)\|_2^2] \leq \sum_{t=0}^{T-1} \mathbb{E}[\|\nabla\Phi(x_t) - \nabla_x f(x_t, y_t) + \nabla_x f(x_t, y_t) - v_t + v_t\|_2^2]$$

$$\leq 3 \sum_{t=0}^{T-1} \left( \mathbb{E}[\|\nabla\Phi(x_t) - \nabla_x f(x_t, y_t)\|_2^2] + \mathbb{E}[\|\nabla_x f(x_t, y_t) - v_t\|_2^2] + \mathbb{E}[\|v_t\|_2^2] \right)$$

$$\leq 3 \sum_{t=0}^{T-1} \left( \kappa^2 \mathbb{E}[\|\nabla_y f(x_t, y_t)\|_2^2] + \mathbb{E}[\|\nabla_x f(x_t, y_t) - v_t\|_2^2] + \mathbb{E}[\|v_t\|_2^2] \right)$$

$$\leq 3\kappa^2 \sum_{t=0}^{T-1} \delta_t + 3 \sum_{t=0}^{T-1} \Delta_t + 3 \sum_{t=0}^{T-1} \mathbb{E}[\|v_t\|_2^2]$$

$$\overset{(i)}{\leq} 6\kappa^2 \sum_{t=0}^{T-1} \delta'_t + 6 \sum_{t=0}^{T-1} \Delta'_t + 3 \sum_{t=0}^{T-1} \mathbb{E}[\|v_t\|_2^2] + 3T\pi(d_1, d_2, \mu_1, \mu_2) \tag{63}$$

where $(i)$ follows from Lemma 16. Substituting eq. (57), eq. (58) and eq. (62) into eq. (63) yields

$$\sum_{t=0}^{T-1} \mathbb{E}[\|\nabla\Phi(x_t)\|_2^2]$$

$$\leq (60\kappa^2 + 24)T\epsilon(S_1, \delta) + (24\kappa^2 + 3)\delta'_0 + (60\kappa^2 + 3)T\pi_\Delta(d_1, d_2, \mu_1, \mu_2)$$

$$+ (24\kappa^2 + 24)T\pi_\delta(d_1, d_2, \mu_1, \mu_2) + (144\kappa^2\alpha^2\ell^2 + 42\alpha^2\ell^2 + 3) \sum_{t=0}^{T-1} \mathbb{E}[\|v_t\|_2^2]$$

$$+ 3T\pi(d_1, d_2, \mu_1, \mu_2)$$

$$\overset{(i)}{\leq} 84\kappa^2 T\epsilon(S_1, \delta) + 27\kappa^2\delta'_0 + 63\kappa^2 T\pi_\Delta(d_1, d_2, \mu_1, \mu_2) + 48\kappa^2 T\pi_\delta(d_1, d_2, \mu_1, \mu_2)$$

$$+ 4 \sum_{t=0}^{T-1} \mathbb{E}[\|v_t\|_2^2] + 3T\pi(d_1, d_2, \mu_1, \mu_2)$$

$$\overset{(ii)}{\leq} 288L(\Phi(x_0) - \Phi^*) + 420\kappa^2 T\epsilon(S_1, \delta) + 135\kappa^2\delta'_0 + 315\kappa^2 T\pi_\Delta(d_1, d_2, \mu_1, \mu_2)$$

$$+ 240\kappa^2 T\pi_\delta(d_1, d_2, \mu_1, \mu_2) + 15T\pi(d_1, d_2, \mu_1, \mu_2). \tag{64}$$

where $(i)$ follows from the fact that $\kappa > 1$, $L = (\kappa + 1)\ell$ and $\alpha = \frac{1}{24L}$, and $(ii)$ follows from eq. (62). Recall $L = (1 + \kappa)\ell$. Then, eq. (64) implies that

$$\mathbb{E}[\|\nabla\Phi(\hat{x})\|_2^2]$$

$$\leq 288(\kappa + 1)\ell \frac{\Phi(x_0) - \Phi^*}{T} + 420\kappa^2\epsilon(S_1, \delta) + \frac{135\kappa^2\delta'_0}{T}$$

$$+ 315\kappa^2 \pi_\Delta(d_1, d_2, \mu_1, \mu_2) + 240\kappa^2 \pi_\delta(d_1, d_2, \mu_1, \mu_2) + 15\pi(d_1, d_2, \mu_1, \mu_2). \tag{65}$$

Recalling Lemma 10, we have

$$\epsilon(S_1, \delta) \leq \frac{(d_1 + d_2)\ell^2\delta^2}{2} + \frac{4\sigma^2}{S_1} + \frac{\mu_1^2}{2}\ell^2(d_1 + 3)^3 + \frac{\mu_2^2}{2}\ell^2(d_2 + 3)^3.$$

If we let $\delta'_0 \le \frac{1}{\kappa}$, $T = \max\{1728(\kappa+1)\ell\frac{\Phi(x_0)-\Phi^*}{\epsilon^2}, \frac{810\kappa}{\epsilon^2}\}$, $S_1 = \frac{40320\sigma^2\kappa^2}{\epsilon^2}$, and further let $\delta = \frac{\epsilon}{71\kappa\ell\sqrt{d_1+d_2}}$, $\mu_1 = \frac{\epsilon}{71\kappa^{2.5}\ell(d_1+6)^{1.5}}$ and $\mu_2 = \frac{\epsilon}{71\kappa^{2.5}\ell(d_2+6)^{1.5}}$, according to the definition of $\epsilon(S_1,\delta)$ (Lemma 10), $\pi_\Delta(d_1,d_2,\mu_1,\mu_2)$ (Lemma 17), $\pi_\delta(d_1,d_2,\mu_1,\mu_2)$ (Lemma 18) and $\pi(d_1,d_2,\mu_1,\mu_2)$ (eq. (48)), then we have $420\kappa^2\epsilon(S_1,\delta) \le \frac{\epsilon^2}{6}$, and

$$315\kappa^2\pi_\Delta(d_1,d_2,\mu_1,\mu_2) + 240\kappa^2\pi_\delta(d_1,d_2,\mu_1,\mu_2) + 15\pi(d_1,d_2,\mu_1,\mu_2) \le \frac{\epsilon^2}{2},$$

which implies

$$\mathbb{E}[\|\nabla\Phi(\hat{x})\|_2] \le \sqrt{\mathbb{E}[\|\nabla\Phi(\hat{x})\|_2^2]} \le \epsilon.$$

We also let $S_{2,x} = \frac{5600(d_1+4)\kappa}{\epsilon}$, $S_{2,y} = \frac{5600(d_2+4)\kappa}{\epsilon}$ and $q = \frac{2800\kappa}{13\epsilon(\kappa+1)}$. Then, the total sample complexity is given by

$$T \cdot (S_{2,x} + S_{2,y}) \cdot m + \left\lceil \frac{T}{q} \right\rceil \cdot S_1 \cdot (d_1+d_2) + T_0$$

$$\le \Theta\left(\frac{\kappa}{\epsilon^2} \cdot \frac{(d_1+d_2)\kappa}{\epsilon} \cdot \kappa\right) + \Theta\left(\frac{\kappa}{\epsilon} \cdot \frac{\kappa^2}{\epsilon^2} \cdot (d_1+d_2)\right) + \Theta\left(d_2\kappa\log(\kappa)\right)$$

$$= \mathcal{O}\left(\frac{(d_1+d_2)\kappa^3}{\epsilon^3}\right),$$

which completes the proof. $\qquad\square$

## E  Proof of Theorem 3

In the finite-sum case, recall that

$$f(x,y) \triangleq \frac{1}{n}\sum_{i=1}^{n} F(x,y;\xi_i).$$

Here we modify Algorithm 1 by replacing the mini-batch update used in line 6 and 7 of Algorithm 1 with the following update using all samples:

$$v_t = \frac{1}{n}\sum_{i=1}^{n}\sum_{j=1}^{d_1} \frac{F(x_t+\delta e_j, y_t, \xi_i) - F(x_t-\delta e_j, y_t, \xi_i)}{2\delta}e_j,$$

$$u_t = \frac{1}{n}\sum_{i=1}^{n}\sum_{j=1}^{d_2} \frac{F(x_t, y_t+\delta e_j, \xi_i) - F(x_t, y_t-\delta e_j, \xi_i)}{2\delta}e_j,$$

where $e_j$ denotes the $j$-th canonical unit basis vector. In this case, if $\text{mod}(k,q) = 0$, then we have

$$\epsilon(S_1,\delta) \le \frac{(d_1+d_2)\ell^2\delta^2}{2} + \frac{\mu_1^2}{2}\ell^2(d_1+3)^3 + \frac{\mu_2^2}{2}\ell^2(d_2+3)^3. \tag{66}$$

**Case 1:** $n \ge \kappa^2$

Substituting eq. (66) into eq. (65), it can be checked easily that under the same parameter settings for $\delta'_0$, $T$, $\delta$, $\mu_1$ and $\mu_2$ in Theorem 2, we have

$$\mathbb{E}[\|\nabla\Phi(\hat{x})\|_2] \le \sqrt{\mathbb{E}[\|\nabla\Phi(\hat{x})\|_2^2]} \le \epsilon.$$

Then, let $S_{2,x} = 5600(d_1+4)\kappa\sqrt{n}$, $S_{2,y} = 5600(d_2+4)\kappa\sqrt{n}$ and $q = \frac{2800\kappa\sqrt{n}}{13(\kappa+1)}$. Recalling the sample complexity result of ZO-iSARAH in the finite-sum case in Appendix C, we have $T_0 = \mathcal{O}\left(d_2(\kappa+n)\log(\kappa)\right)$. The total sample complexity is given by

$$T \cdot (S_{2,x} + S_{2,y}) \cdot m + \left\lceil \frac{T}{q} \right\rceil \cdot S_1 \cdot (d_1+d_2) + T_0$$

$$\leq \Theta\left(\frac{\kappa}{\epsilon^2} \cdot (d_1 + d_2)\sqrt{n} \cdot \kappa\right) + \Theta\left(\left\lceil\frac{\kappa^2}{\sqrt{n}\epsilon^2}\right\rceil \cdot n \cdot (d_1 + d_2)\right) + \Theta\left(d_2(\kappa + n)\kappa\log(\kappa)\right)$$

$$= \mathcal{O}\left((d_1 + d_2)(\sqrt{n}\kappa^2\epsilon^{-2} + n)\right) + \mathcal{O}(d_2(\kappa^2 + \kappa n)\log(\kappa)).$$

**Case 2:** $n \leq \kappa^2$

In this case, we let $S_{2,x} = 56(d_1 + 4) + 420$, $S_{2,y} = 56(d_2 + 4) + 420$ and $q = 1$. Then we have

$$\Delta'_t \leq \epsilon_\Delta = \frac{(d_1 + d_2)\ell^2\delta^2}{2} + \frac{\mu_1^2}{2}\ell^2(d_1 + 3)^3 + \frac{\mu_2^2}{2}\ell^2(d_2 + 3)^3, \quad \text{for all} \quad 0 \leq t \leq T - 1. \tag{67}$$

Given the value of $S_{2,x}$ and $S_{2,y}$, it can be checked that the proofs of Lemma 15 and Lemma 18 still hold. Following from the steps similar to those from eq. (47) to eq. (56), we obtain

$$\sum_{t=0}^{T-1} \delta'_t \leq 2\delta'_0 + \frac{5}{2}T\epsilon_\Delta + 6\ell^2\alpha^2\sum_{t=0}^{T-2}\mathbb{E}[\|v_t\|_2^2] + 2T\pi_\delta(d_1, d_2, \mu_1, \mu_2). \tag{68}$$

Substituting eq. (67) and eq. (68) into eq. (47) yields

$$\left(\frac{\alpha}{2} - \frac{L\alpha^2}{2}\right)\sum_{t=0}^{T-1}\mathbb{E}[\|v_t\|_2^2]$$

$$\overset{(i)}{\leq} \Phi(x_0) - \mathbb{E}[\Phi(x_T)] + 4\alpha\kappa^2\delta'_0 + 7\alpha\kappa^2 T\epsilon_\Delta + 12L^2\alpha^3\sum_{t=0}^{T-2}\mathbb{E}[\|v_t\|_2^2]$$

$$+ 4\alpha\kappa^2 T\pi_\delta(d_1, d_2, \mu_1, \mu_2) + \alpha T\pi(d_1, d_2, \mu_1, \mu_2), \tag{69}$$

where $(i)$ follows because $L = (1 + \kappa)\ell$. Rearranging eq. (69) yields

$$\left(\frac{\alpha}{2} - \frac{L\alpha^2}{2} - 12L^2\alpha^3\right)\sum_{t=0}^{T-1}\mathbb{E}[\|v_t\|_2^2]$$

$$\leq \Phi(x_0) - \mathbb{E}[\Phi(x_T)] + 4\alpha\kappa^2\delta'_0 + 7\alpha\kappa^2 T\epsilon_\Delta + 4\alpha\kappa^2 T\pi_\delta(d_1, d_2, \mu_1, \mu_2) + \alpha T\pi(d_1, d_2, \mu_1, \mu_2). \tag{70}$$

Letting $\alpha = \frac{1}{8L}$, we obtain

$$\frac{\alpha}{2} - \frac{L\alpha^2}{2} - 12L^2\alpha^3 = \frac{1}{32L}. \tag{71}$$

Substituting eq. (70) into eq. (71) and applying Assumption 1 yield

$$\sum_{t=0}^{T-1}\mathbb{E}[\|v_t\|_2^2] \leq 32L(\Phi(x_0) - \Phi^*) + 16\kappa^2\delta'_0 + 28\kappa^2 T\epsilon_\Delta + 16\kappa^2 T\pi_\delta(d_1, d_2, \mu_1, \mu_2)$$

$$+ 4T\pi(d_1, d_2, \mu_1, \mu_2). \tag{72}$$

Substituting eq. (72) and eq. (67) into eq. (63) yields

$$\sum_{t=0}^{T-1}\mathbb{E}[\|\nabla\Phi(x_t)\|_2^2]$$

$$\leq 6\kappa^2\sum_{t=0}^{T-1}\delta'_t + 6T\epsilon_\Delta + 3\sum_{t=0}^{T-1}\mathbb{E}[\|v_t\|_2^2] + 3T\pi(d_1, d_2, \mu_1, \mu_2)$$

$$\leq 12\kappa^2\delta'_0 + 21\kappa^2 T\epsilon_\Delta + 4\sum_{t=0}^{T-1}\mathbb{E}[\|v_t\|_2^2] + 12\kappa^2 T\pi_\delta(d_1, d_2, \mu_1, \mu_2) + 3T\pi(d_1, d_2, \mu_1, \mu_2)$$

$$\leq 128L(\Phi(x_0) - \Phi^*) + 76\kappa^2\delta_0' + 133\kappa^2 T\epsilon_\Delta + 76\kappa^2 T\pi_\delta(d_1, d_2, \mu_1, \mu_2) + 19T\pi(d_1, d_2, \mu_1, \mu_2). \tag{73}$$

Recall that $L = (1 + \kappa)\ell$. Then, eq. (73) implies

$$\mathbb{E}[\|\nabla\Phi(\hat{x})\|_2^2] \leq 128(\kappa + 1)\ell\frac{\Phi(x_0) - \Phi^*}{T} + 133\kappa^2\epsilon_\Delta + \frac{76\kappa^2\delta_0'}{T} + 76\kappa^2\pi_\delta(d_1, d_2, \mu_1, \mu_2)$$
$$+ 19\pi(d_1, d_2, \mu_1, \mu_2).$$

If we let $\delta_0' \leq \frac{1}{\kappa}$, $T = \max\{640(\kappa + 1)\ell\frac{\Phi(x_0) - \Phi^*}{\epsilon^2}, \frac{380\kappa}{\epsilon^2}\}$, and let $\mu_1$, $\mu_2$ and $\delta$ follow the same setting in Theorem 2, then we have

$$\mathbb{E}[\|\nabla\Phi(\hat{x})\|_2] \leq \sqrt{\mathbb{E}[\|\nabla\Phi(\hat{x})\|_2^2]} \leq \epsilon.$$

Recall the sample complexity result of ZO-iSARAH in the finite-sum case in Appendix C. Then, we have $T_0 = \mathcal{O}\left(d_2(\kappa + n)\log(\kappa)\right)$. The total sample complexity is given by

$$T \cdot (S_{2,x} + S_{2,y}) \cdot m + \left\lceil\frac{T}{q}\right\rceil \cdot S_1 \cdot (d_1 + d_2) + T_0$$
$$\leq \Theta\left(\frac{\kappa}{\epsilon^2} \cdot (d_1 + d_2) \cdot \kappa\right) + \Theta\left(\left\lceil\frac{\kappa}{\epsilon^2}\right\rceil \cdot n \cdot (d_1 + d_2)\right) + \Theta\left(d_2(\kappa + n)\log(\kappa)\right)$$
$$= \mathcal{O}\left((d_1 + d_2)(\kappa^2 + \kappa n)\epsilon^{-2}\right).$$

