# OpenReview forum: "Gradient-Free Minimax Optimization: Variance Reduction and Faster Convergence"
_TMLR — Rejected by TMLR_

### Review · Reviewer_c92W · 2022-09-19

**Summary Of Contributions:**

This paper studies gradient-free stochastic algorithms for nonconvex-strongly-concave minimax stochastic optimization problem. The authors use the variance reduction technique to design the ZO-VRGDA algorithm, which achieves the best known query complexity of $\mathcal{O}(d\kappa^3\epsilon^{-3})$. The experiments also show the proposed algorithm performs well.

**Broader Impact Concerns:**

This is an theirtical paper. There is no concerns on the ethical implications.

**Requested Changes:**

Major Changes:

As I mentioned in "Weaknesses" section, the DRO problem has a simplex constraint on $y$ while ZO-iSARAH only consider unconstraint case. I think there are two options to improve this paper:
1. Conduct the projection step for the update of $y$ in ZO-VRGDA and provide the corresponding convergence analysis. I believe the total complexity of $\mathcal{O}(d\kappa^3\epsilon^{-3})$ still holds by such modification.
2. If the authors will not modify the convergence analysis a lot, I think it is neccesary to conduct an unconstraint problem in experimental section (for example, the model of domain-adversarial with a simple neural network [1], which is also used for the study of nonconvex-strongly-concave optimization recently [2]). Then it is OK to view the ZO-iSARAH for constrained case as an extension without therotical analysis.

If one of above points has been finished, I will recommend acceptence.

[1] Yaroslav Ganin, Evgeniya Ustinova, Hana Ajakan, Pascal Germain, Hugo Larochelle, Francois Laviolette, Mario Marchand, and Victor Lempitsky. Domain-adversarial training of neural networks. Journal of Machine Learning Research, 17(1):2096–2030, 2016.
[2] Luo Luo, Yujun Li, Cheng Chen. Finding Second-Order Stationary Point for Nonconvex-Strongly-Concave Minimax Problem. arXiv:2110.04814


Minor Changes:
1. Page 2 says "the lower bound on the computational complexity suggests that zeroth-order algorithms may potentially achieve
the query complexity of $\mathcal{O}((d_1+d_2)\epsilon^{-3})$". The lower bound in this sentence in unclear. It is neccesary to provide some reference and discuss the lower bound in details.
2. The abstract says "the ZO-VRGDA algorithm achieves the best known query complexity of  $\mathcal{O}(\kappa(d_1+d_2)\epsilon^{-3})$". It seems to be  $\mathcal{O}(\kappa^3(d_1+d_2)\epsilon^{-3})$.
3. The curves in Figure 1 should be presented by different types of makers.

**Strengths And Weaknesses:**

Strengths

This paper is well written. The deisgn of ZO-VRGDA is well-motivated and intuitive. The complexity analysis in this paper extend the idea of SREAD to gradient-free setting, which is easy to follow. Additionally, it is interseting that ZO-iSARAH has better upper bound than the initializaiton of SREDA in the setting of first-order optimization.

Weaknesses
1. The main consideration is the expeirments. The distributionally robust optimization (DRO) problem has a simplex constraint on $y$, but the exisiting algorithm only address the unconstraint case.
2. Some details of presentation could be improved. Please see "Requested Changes".

---

### Review · Reviewer_Ln2r · 2022-09-29

**Summary Of Contributions:**

This work provides a zero-th order method to approximate stationary points in nonconvex strongly-concave minimax optimization problems. The algorithm is based on (Gaussian) stochastic smoothing to build variance-reduced gradient estimators, which can be plugged into a gradient method, whose gradients are obtained by approximately solving the inner strongly concave problem through another stochastic variance-reduced method. The results apply similarly to stochastic and finite-sum settings. Some numerical experiments (which I couldn't quite understand their value; see below) are also provided.



**Requested Changes:**

1. Page 2. It is mentioned the existence of lower bounds for this problem. Can the authors add a specific reference? If they are referring to the one for first-order methods, perhaps this reference is useful (https://arxiv.org/pdf/2103.15888.pdf). Either way, I don't see clearly how the claimed lower bound holds, so this deserves a more thorough discussion.
2. Page 5. The iSARAH algorithm is discussed, but no reference is provided.
3. Page 6, Algorithm 2. It is not made explicit what function is P?
4. Page 6, Algorithm 3. I don't see where in the algorithm $\tilde v_{t,0}$ or $\tilde v_{t,k}$ are being used. If they are not used, why are they even computed?
5. Page 6, Algorithm 3. Why is the output y defined as a random iterate? This being a stochastic strongly convex program this algorithm should have guarantees for the last iterate.
6. Page 7. I couldn't get why the initialization requirement is based on $\mathbb{E}[\|\nabla_y f(x_0,y_0)\|^2]$. Perhaps the authors can elaborate more on that requirement.
7. Page 10, Experiments, Figure 1. The figure contains convergence results for three algorithms (ZO-SGDA, ZO-SGDMSA, ZO-SREDA-Boost). None of them is the algorithm studied in the paper (ZO-VRGDA). I don't understand why is this done at all.

**Strengths And Weaknesses:**

Strengths:
1. The problem of nonconvex minimax optimization has attracted much attention, and zero-th order methods here seem scarce.
2. The idea of coupling the different errors arising in the algorithm seems interesting. I am not sure whether this is the first work taking such an approach, but if this is the case this should be highlighted.

Weaknesses:
1. Given the existing work on stochastic first-order methods for these problems and Gaussian smoothing, the contributions are somewhat limited.
2. I found parts of the algorithms' descriptions to be confusing. This is possibly a misunderstanding, so hopefully, the authors can provide more details.
3. Similarly to the algorithms' descriptions, I also found the numerical results confusing.

---

### Review · Reviewer_V9Y2 · 2022-10-23

**Summary Of Contributions:**

The paper introduces a variance-reduced zero-order algorithm built upon SREDA for nonconvex-strongly-concave minimax optimization. Moreover, because of SPIDER-type variance reduction, it improves over the previous zero-order methods in this setting by a factor of $\epsilon^{-1}$.

**Broader Impact Concerns:**

The paper has no specific ethical implications.

**Requested Changes:**

1.  In Page 2, please elaborate why $O((d_1 + d_2)\epsilon^{-3})$ lower complexity bound for first-order method is also meaningful for zero-order setting in this paper, and add reference for this complexity bound.
2. [minor] In Section 2.1, in the first sentence under the equation, it should be $\nu$ and $\omega$ instead of $\nu_i$ and $\omega_i$
3. For Assumption 3, I suggest to comment on requirement of strong-concavity for each component function.
4. [minor] In the algorithm box of Algorithm 1, in line 10, $\bar{m}$ is never defined. I think it should be $\tilde{m}$ and please mention it is from Algorithm 3. In line 4, it should be "t" instead of "k".
5. Although the algorithm stems from SREDA, I still suggest to explain why different variance-reduction estimators are used in Algorithm 1 and 3, e.g., the difference in the condition of $x$ and $y$.
6. [minor] The value of $\tilde{x}_{t, k}$ seems to be unchanged in Algorithm 3. Why is it necessary to include step 11?
7. [minor] The algorithm name in legend of Figure 1 is wrong.
8. In experiment, Figure 1 and Figure 2, these algorithm are run for different number of queries and it is hard to compare them now. I suggest to run them for a fixed number of queries. Also, why do the curves in Figure 1(b) and Figure 2(b) look quite different?
9. [minor] Page 20, the line below (27), the inequality (iii) is not marked.

**Strengths And Weaknesses:**

### Strengths
1. It is the first zero-order algorithm for this setting that achieves $\epsilon^{-3}$ complexity.
2. The overall structure of the paper is well-organized.
3. The method in this paper does not depend on $\epsilon$ and does not require high accuracy initialization.

### Weaknesses
1. The result is not surprising, since $\epsilon^{-3}$ complexity is already established for that first-order method SREDA, and $\epsilon$-independent stepsize and inaccurate initialization are used in SPIDER-boost for nonconvex optimizatoin.
2. The writing needs to be improved and there are some typos. For example, the complexity in the abstract is not correct: $\kappa$ -> $\kappa^3$.
3. Note that $\epsilon$-independent stepsize can not be easily compared to the stepsize in SREDA, because SREDA actually uses a stepsize normalized by $||v_t||$ and the stepsize can be large when $||v_t||$ is small.
4. The problem is not fully motivated. In the introduction, the paper mentioned that some examples in RL do not have access to first-order information. I wonder whether in those applications, the zero-order estimates of gradient in the form (3) (4) can be easily attained with large-batch size.
5. I am not sure whether experiments are fair, because batch size in ZO-VRGDA is chosen to be n instead of the batch size what suggested in theory. There are many parameters to tune in each algorithm, and the desirable constant C may be different for each of them. Even if using the same constant C for all of them, I suggest to report for different choices of C.

---

### Review · Reviewer_ku2U · 2022-10-24

**Summary Of Contributions:**

This work aims at solving nonconvex-strongly-concave minimax stochastic optimization problems using only function value oracle, known as zeroth-order or gradient-free optimization. The author proposed a novel variance-reduced gradient descent-ascent algorithm (ZO-VRGDA) that achieves the best-known query complexity of $\mathcal{O}\left((d_1+d_2)\epsilon^{-3}\right)$ in the expectation setting, and $\mathcal{O}\left((d_1+d_2)n^{1/2}\epsilon^{-2}\right)$ in the finite-sum setting (leading-order term when $n\gg 1$ and $\kappa = O(1)$), outperforming all existing stochastic algorithm complexity bounds by orders of magnitude, where $d_1+d_2$ denote the sum of variable dimensions and $\kappa$ denotes the condition number. The work is further enriched by simulation results of black-box distributional robust optimization, demonstrating the outperformance of their proposed algorithm.


**Broader Impact Concerns:**

No concerns due to the theoretical/mathematical nature of this work.

**Requested Changes:**

Page 2, section 1.1: "summary of contributions" is unnecessarily lengthy and relatively pointless. I suggest the authors itemize them with fewer discussion texts (and move additional texts into their corresponding later sections).

Page 5, Lemma 1: typographical error when citing Lin et al.’s lemma, $y^*$ is $\kappa$-Lipschitz instead of $\nabla \Phi(\cdot)$ placed here.

Page 6, first line: “where $\psi_i\sim N(0, \mathbf{1}_d)$" should be $\psi_i\sim N(0, \mathbf{I}_d)$.

Page 8, before Section 4.2: “... Theorem 3 provides the first query complexity for the finite-sum zeroth-order minimax problems”. This kind of “first” expression may result in unnecessary false claims. In fact, the case of $n=1$ can easily extend to a general finite-sum setting with a multiplicative factor $n$ of individual function number in its complexity bound, e.g. Wang et al. (2020), Liu et al. (2019).

Page 10, Figure 1: legend part missing a label of ZO-VRGDA, and there is a new label of ZO-SREDA-Boost. Are they different names for the same algorithm?

Page 11, references: some references might be valuable to be added: [1] achieves the $n^{1/2}\epsilon^{-2} \land \epsilon^{-3}$ compelxity as a concurrent work of Fang et al. (2018). [2] achieves a sharper rate for nonconvex-concave first-order minimax optimization than Lin et al. (2019). [3] is regarded as the first in literature prevalent variance-reduced stochastic gradient method for strongly convex objectives. [4] provides a single-loop stochastic nonconvex optimization method that achieves an optimal $O(\epsilon^{-3})$ complexity up to a problem-dependent factor. [5] and [6] provide $\Omega(\epsilon^{-3})$ lower bound results for the analogous first-order methods.

I also suggest that the authors conduct a more thorough bibliographical review and add other significant references that are closely related to this work.

[1] Dongruo Zhou, Pan Xu, and Quanquan Gu. "Stochastic nested variance reduction for nonconvex optimization." In NeurIPS, 2018.

[2] Tianyi Lin, Chi Jin, and Michael I. Jordan. "Near-optimal algorithms for minimax optimization." Conference on Learning Theory. PMLR, 2020.

[3] Nicolas Le Roux, Mark Schmidt, and Francis Bach. "A stochastic gradient method with an exponential convergence rate for finite training sets." Advances in neural information processing systems 25 (2012).

[4] Ashok Cutkosky and Francesco Orabona. "Momentum-based variance reduction in non-convex sgd." Advances in neural information processing systems 32 (2019).

[5] Yossi Arjevani, Yair Carmon, John C. Duchi, Dylan J. Foster, Nathan Srebro, and Blake Woodworth. "Lower Bounds for Non-Convex Stochastic Optimization." arXiv preprint arXiv:1912.02365 (2019).

[6] Siqi Zhang, Junchi Yang, Cristóbal Guzmán, Negar Kiyavash, and Niao He. "The complexity of nonconvex-strongly-concave minimax optimization." In Uncertainty in Artificial Intelligence, pp. 482-492. PMLR, 2021.

**Strengths And Weaknesses:**

STRENGTH
--The complexity enjoys the best known complexity upper bound of $O((d_1+d_2)\epsilon^{-3})$, which matches the suggested complexity by the lower bound of gradient-based algorithms. Indeed, this is one of the best results in this regime [Theorems 1 \& 2]. The writing is generally clear and readers with proper training in stochastic optimization (min-only) can easily follow.


WEAKNESS
--Technical novelty might be limited. The analysis in this work is largely based on existing recursive variance-reduced methods for minimization (e.g., Fang et al. (2018), Wang et al. (2019), Nguyen et al. (2018)) and minimax optimization (e.g., Luo et al. (2020)). The authors claimed that they have adopted accuracy-independent stepsizes and initial batchsizes over earlier results while maintaining the theoretical convergence guarantee---this avoids the necessity of making both tracking and gradient-estimation errors to be constantly $O(\epsilon)$. However, this is believed to be already a commonly seen technique in algorithmic analyses, especially for recursive variance-reduced methods (e.g., Wang et al. (2019)).

--This might not be categorized as weakness, but the zeroth-order approximation of first-order oracle is in fact a mix of coordinate-wise estimators (outer) and zeroth-order Gaussian smooth estimators (inner) as estimators of gradients, which seems unnatural. What are the features/differences between these two estimators? Can we adopt solely one of these estimators to obtain desirable results?

--Experiments part seems not as exhaustive as one might expect. Comparisons on the performance of zeroth-order algorithms have been made with only ZO-SGDA and ZO-SGDMSA (Wang et al., 2020) designed for nonconvex-strongly-concave minimax problems for three datasets from LIBSVM. Seems there are many methods that are omitted, including but not only Acc-ZODMA in Huang et al. (2020). More cases for synthetic and real datasets should also be explored.

---

### Decision · Action_Editors · 2022-11-10

**Recommendation:** Reject

**Comment:**

In this paper, the authors consider nonconvex-strongly-concave minimax stochastic optimization problem, and focus on the gradient-free setting. Based on the variance reduction technique, they design a novel zeroth-order variance reduced gradient descent ascent (ZO-VRGDA) algorithm, and establish a better query complexity.

The reviewers have serious concerns regarding the technical novelty, the experimental setting and the writing. In particular, the authors are requested to clarify some details in the analysis, and conduct more experiments. However, they are unable to revise this submission before the deadline of response. Because we do not have the option of major revision, I decide to reject it now, and encourage the authors to submit a significantly revised version later.

**Audience:**

Yes.

**Claims And Evidence:**

Yes. But according to the reviews, the authors need to improve both the theoretical analysis and empirical studies.